

# Assessing bias-corrections of oceanic surface conditions for atmospheric models

Julien Beaumet[1], Gerhard Krinner[1], Michel Déqué[2], Rein Haarsma[3], and Laurent Li[4]

[1]Univ. Grenoble Alpes, CNRS, IRD, Institut des Géosciences de l'Environnement, F-38000, Grenoble, France
[2]Météo-France, Centre National de Recherche Météorologiques, Toulouse, France
[3]Royal Netherlands Meteorological Institute (KNMI), De Bilt, Netherlands
[4]Laboratoire de Météorologie Dynamique, Université Pierre-Marie Curie, CNRS, Paris, France

*Correspondence to:* Julien Beaumet (Julien.Beaumet@univ-grenoble-alpes.fr)

**Abstract.** Future sea-surface temperature and sea-ice concentration from coupled ocean-atmosphere general circulation models such as those from the CMIP5 experiment are often used as boundary forcing for the downscaling of future climate experiment. Yet, these models show some considerable biases when compared to the observations over present climate. In this paper, existing methods such as an absolute anomaly and a quantile-quantile method for sea surface temperature (SST) as well as a

look-up table and a relative anomaly method for sea-ice concentration (SIC) are presented. For SIC, we also propose a new analog method. Each method is objectively evaluated with a perfect model test using CMIP5 model experiment and some real-case applications using observations. With respect to other previously existing methods for SIC, the analog method is a substantial improvement for the bias correction of future sea-ice concentrations.

*Copyright statement.* TEXT

**1 Introduction-Context**

Coupled climate models are the most reliable tools that we have today for large-scale climate projections, such as in the CMIP5 model intercomparison project (Taylor et al., 2012), in which these projections were based on representative concentration pathways (RCPs; Moss et al. (2010)). Regional-scale information is obtained by using these global simulations as a basis for downscaling exercises. Dynamical downscaling, as opposed to empirical-statistical downscaling (e.g., Hewitson et al.,

2014), is carried out either with Regional Climate Models (e.g., Giorgi and Gutowski, 2016) or with high-resolution global atmospheric general circulation models (Haarsma et al., 2016). In both cases, information about the projected changes of sea-surface conditions (sea-surface temperatures (SST) and sea-ice concentration (SIC)) is required as a lower boundary condition for the higher-resolution models. However, SST and SIC conditions modelled by coupled AOGCMs show important biases for the present climate (Flato et al., 2013). It has, for example, been highlighted that most of the CMIP5 models had difficulties

in reliably modelling seasonal cycle and trend of sea-ice extent in the Antarctic over the historical period (Turner et al., 2013). Therefore, the validity and reliability of such coupled simulations is questionable for future climate projections (e.g. end of the





21st century) and so is their use as boundary conditions when performing dynamical downscaling of future climate projections. Prescribing correct SST is crucial for atmospheric modelling because SST determines heat and moisture exchanges with the atmosphere (Ashfaq et al., 2011; Hernández-Díaz et al., 2017). In high latitude, sea-ice concentration (Krinner et al., 2008; Screen and Simmonds, 2010; Noël et al., 2014) and sea-ice thickness (Gerdes, 2006; Krinner et al., 2010) are two additional

required and crucial boundary conditions for atmospheric modelling of recent and future climate change. Krinner et al. (2014) demonstrated that for the Antarctic climate as simulated by an atmospheric model, prescribed SST and sea-ice changes have greater influence than prescribed greenhouse gas concentration changes. Integrated winter sea-ice extent and summer SST have been identified among the key boundary forcings for regional modelling of the Antarctic surface mass balance (Agosta et al., 2013), which is the only potentially significant negative contributor to the global eustatic sea level change in the course of the

21st century (Agosta et al., 2013; Church et al., 2013; Lenaerts et al., 2016).

In this study, we describe, evaluate and discuss different existing and new methods for the construction of bias-corrected future SST and SIC. These methods generally take into account observed oceanic boundary conditions as well as the climate change signal coming from CMIP5 AOGCM scenarios to build more reliable SST and SIC conditions for future climate, which should reduce the uncertainties when used to force future climate projections. The different methods have been evaluated using a

perfect model test and by carrying out real-case applications on observations. Applied changes in mean and variances have been investigated as well as the coherence of SIC and SST after applying bias correction methods. The analysis of the results focuses on methods for sea-ice, as bias correction of SIC is a more complicated issue to deal with. In the following, we present the bias-correction methods, the data and the evaluation methods in section 2. The results of the evaluation are shown in section 3 and are discussed together with general considerations on bias correction of oceanic surface conditions in section 4. Finally,

our findings are summed up and we draw conclusions in section 5.

## 2 Data and methods

### 2.1 Data

Application and validation of the methods for bias correction have been achieved using PCMDI observed SST and SIC that are generally used as boundary conditions for AMIP experiments (Taylor et al., 2000), called "PCMDI obs." or "observations"

in this paper. The AOGCM's historical and future simulated sea-surface conditions come from CMIP5 simulations (Taylor et al., 2012) . Only the first ensemble members of the historical, rcp4.5 and rcp8.5 simulations have been considered. Most methods have been tested using CNRM-CM5, IPSL-CM5A-LR and HadGEM-ES coupled GCM. Data from NorESM1-M and the CCSM4 models have also been used as analog candidates in the analog method for sea-ice. Prior to any application of the bias correction methods, AOGCM data have been bilinearly regridded onto a common regular 1°x1° grid.



## 2.2 Sea Surface Temperature methods

The bias correction of simulated SST is a fairly easy and a straightforward issue to deal with. Nevertheless, different methods have been developed. In this section, we describe an anomaly-based method and a quantile-quantile method. Results from their application are presented in section 3.

### 2.2.1 Anomaly method

This frequently used method (e.g., Krinner et al., 2008) simply consists in adding the SST anomaly coming from the difference between a coupled AOGCM projection and the corresponding historical simulation to the present-day observations. In practice, for each grid point, the difference between the SST for a given month in the future from a climate change simulation and the climatological mean SST in the corresponding historical simulation from the same coupled AOGCM is added to the observed climatological mean SST (e.g . PCMDI, 1971-2000):

$$SST_{Fut,est} = SST_{obs} + (SST_{Fut,AOGCM} - SST_{Hist,AOGCM}) \tag{1}$$

In (1), $SST_{\text{Fut,est}}$ is the estimated future SST for a given month, $SST_{\text{obs}}$ the observed climatological monthly mean, $SST_{\text{Fut,AOGCM}}$ the model future SST for a given month in the future AOGCM scenario and $SST_{\text{Hist,AOGCM}}$ the model climatological monthly mean in the AOGCM historical simulation for the same reference period as for the observed climatology. As a result, the reconstructed SST time series has the chronology of the AOGCM future scenario.

### 2.2.2 Quantile-quantile method

This method has been proposed and described in Ashfaq et al. (2011) It consists in adding, for each grid point and each calendar month's quantile in the observations, the corresponding quantile change in the GCM data set, i.e. the difference between the maximum SST in the future scenario and in the historical simulation, between the second highest SSTs in the two simulations, and so on for each ranked SST quantile. However, unlike Ashfaq et al. (2011), we did not create a new SST field for the present by replacing SST from the GCM in the historical period by its corresponding quantile in the observations, but we directly added the quantile change to the corresponding quantile of the observational time series (Figure 1). This allows keeping the observations chronology and their inter-annual variability in estimated SSTs for the future. In our results, we noticed a large fine-scale spatial variability of the constructed future SSTs that was due to the large spatial variability of the climate change increments (quantile change) calculated individually for each pixel. To fix this, we applied a slight spatial filtering (3 grid point Hann box filter) of the quantile shifts in order to produce more consistent SST fields.

## 2.3 Sea-ice Concentration methods

Sea-ice concentration is more difficult to bias correct because it is a relative quantity that must be strictly bounded between 0 and 100 %. In this section, we present three methods: a look-up table, a relative anomaly and an analog method.





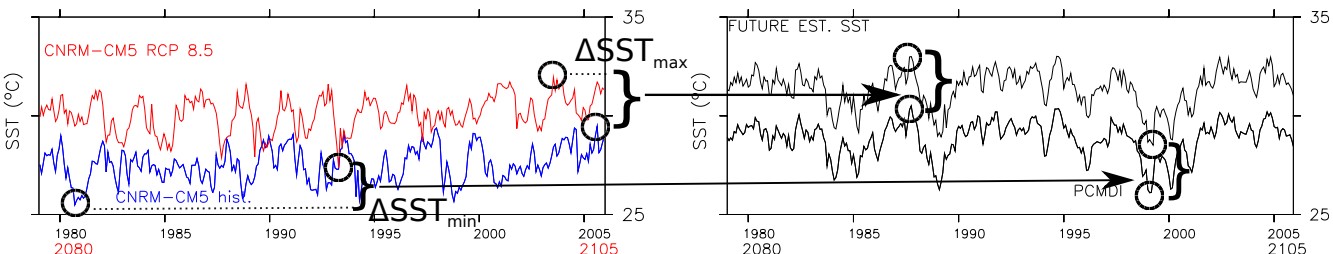

**Figure 1.** Illustration of the quantile-quantile method for min. and max. of SST time series for a grid point in the Central Pacific : GCM historical simulation (blue, left), GCM future scenario (red, left), observed SST(thin, right), reconstructed future SST (thick, right)

### 2.3.1 Look-up Table method

This method has been developed at *the Royal Netherlands Meteorological Institute* (KNMI). It is used in Haarsma et al. (2013) and within the framework of HighResMIP (Haarsma et al., 2016). It is based on the assumption that SIC is a function of SST. Therefore, SST are ranked per 0.1 K bins and the corresponding average SIC for each temperature bin between -2 and +5°C

5   is calculated. Relations between SST and SIC have been found to be dependent on seasons and hemispheres. Therefore, using monthly mean values of SST and SIC from historical observations, look-up tables are built, separately for the Arctic and the Antarctic, for each calendar month (Figure 2). Then, with the help of future SSTs, these look-up tables (LUT) are used to retrieve future SIC.

### 2.3.2 Iterative relative anomaly method

Here we follow a method described by Krinner et al. (2008) based on relative regional sea-ice area (SIA) changes which is essentially an iterative scheme of mathematical morphology for image erosion and dilation (Haralick et al., 1987). The Arctic and the Antarctic are divided into sectors of equal longitude. In each sector, the average SIA is calculated by spatially integrating SIC. With respect to the method introduced in Krinner et al. (2008), we introduce the use of a quantile-quantile

15   method to determine the future targeted SIA. The targeted future SIA is then calculated for each sector and each quantile, with the help of the following equation:

$$SIA_{Fut,est} = SIA_{obs} \cdot \left( \frac{SIA_{Fut,AOGCM}}{SIA_{Hist,AOGCM}} \right) \tag{2}$$

In (2), $SIA_{\text{Fut,est}}$ is the estimated sea ice area for the future for the current month and sector, $SIA_{\text{Obs}}$ the sea-ice area from the observations, and $SIA_{\text{Fut,AOGCM}}$ and $SIA_{\text{Hist,AOGCM}}$ are respectively computed SIA for the corresponding quantile to the

20   observations, using SIC from a future scenario and a historical AOGCM's simulation. Starting from an observed present SIC map and using the computed relative SIA change for a given sector, the decrease (increase) in SIC is then realized using an iterative process: SIC in each grid box is replaced by the minimum (maximum) SIC of all adjacent pixels (Figure 3); the new spatially integrated SIA is calculated and the operation is repeated until the obtained change converges towards the computed





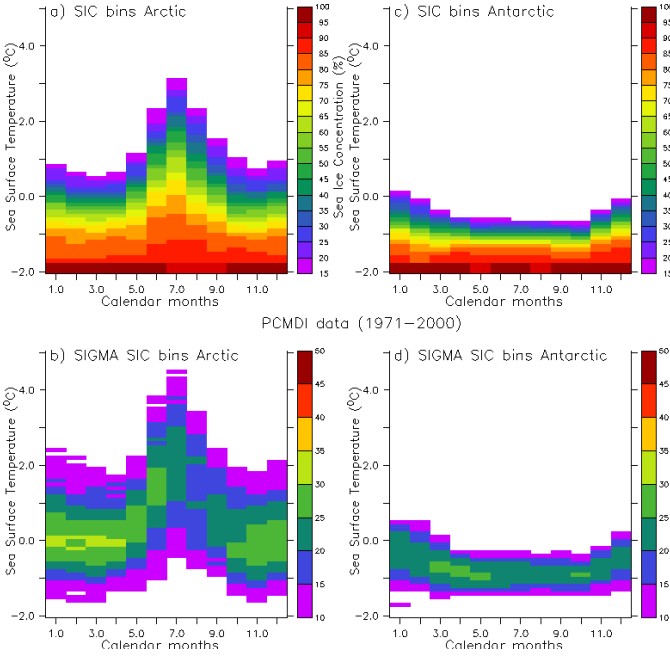

**Figure 2.** Look Up Tables (top) linking SST and SIC for the Arctic (left) and the Antarctic (right) built using 1971-2000 PCMDI observations and the associated uncertainty (root mean square error) on the computed SIC average (bottom)

targeted SIA retrieved from AOGCM's simulation sea-ice data and observations. Afterwards, the decrease/increase process is repeated on the hemisphere scale in order to ensure that the change in SIC reproduces the total hemispheric SIA change.

### 2.3.3 Analog method

In this method, we divide the Arctic and the Antarctic into $n_s$ geographical sectors that correspond to different seas of the
5   Arctic and the Southern Oceans; we defined $n_s$ = 12 sectors for the Arctic and $n_s$ = 7 sectors for the Antarctic. For each sector and each month, the quantiles of the sea-ice extent (SIE: total area with SIC above 15%) and the sea-ice area (SIA) are computed from SIC observations over the AMIP period. Corresponding quantile changes in SIE and SIA are computed using SICs from a CMIP5 AOGCM's historical simulation and future scenario run. Computed quantile changes are then applied to the corresponding quantiles in the observations in order to obtain target future sea-ice extents and areas for each month, quantile
10  and sectors. Then, a library of future SIC fields is built by collecting SIC observations from the AMIP period as well as SIC from CMIP5 projections. The presence of SIC maps from futures AOGCM projections in this library is justified by the need to take into account physically plausible future SIC distributions outside of the current observed range. However, AOGCM that overly poorly represent sea-ice distribution in present-day climate are preferably dismissed from this library. Future SIC is then finally reconstructed by searching the analog for each quantile $q$, sector $s$ and month $m$ in the library, that is to say the SIC field



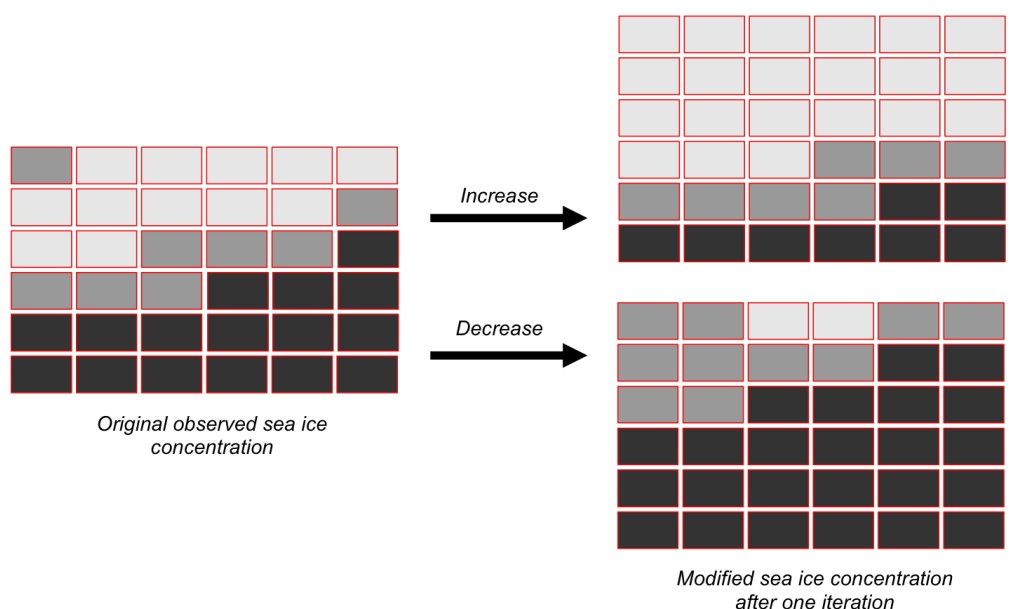

**Figure 3.** Iteratively constructing a "corrected" future SIC field using the iterative relative anomaly method (see text)

that minimizes the cost function C expressed by:

$$C_{q,m,s} = \sqrt{\left(\frac{SIA_s - SIA_{T_{(q,m,s)}}}{SIA_{max_{(q,m,s)}}}\right)^2 + \left(\frac{SIE_s - SIE_{T_{(q,m,s)}}}{SIE_{max_{(q,m,s)}}}\right)^2} \tag{3}$$

where $SIA_s$ and $SIE_s$ are the SIA and SIE of the processed sectors of the analog candidate from the library, $SIA_{T_{(q,m,s)}}$ and

5 $SIE_{T_{(q,m,s)}}$ are the targeted future sea-ice area and extent computed using the quantile-quantile method, and $SIA_{max_{(q,m,s)}}$ and $SIE_{max_{(q,m,s)}}$ are the maximum SIA and SIE of the processed sector. The double criterion on both SIE and SIA was introduced in order to be able to distinguish cases in which the total sea-ice extent in a sector is similar but the average SIC is very different (and vice versa). In order to avoid issues introduced by different land masks between AOGCMs and PCMDI data, we filled land grid points with sea-ice using a nearest neighbour method and masked all the grid points with the same land mask built

with land fraction from PCMDI data in order to compute SIEs and SIAs for each region with the same reference. Analogs are attributed without taking into account the month of the analog candidate in the library. This allows for instance attributing a summer sea-ice map from present observations for a future winter month reconstructed sea-ice field. For each quantile $q$, month $m$ and sector $s$, this procedure yields an hemispheric sea-ice concentration field $SIC_{opt_{(i,q,m,s)}}$ that minimizes the cost function for the given sector, month and quantile. For a given month and quantile, there are thus $n_s$ hemispheric SIC fields

$SIC_{opt_{(i,q,m,s)}}$. At each grid point i, the corresponding $n_s$ SIC values are then blended using a weight function $w_{(i,s)}$ depending on the distance $d_{(i,s)}$ of that grid point to the center of each of the sectors in order to obtain the final reconstructed sea-ice



concentration $SIC_{(i,q,m)}$ for a given quantile $q$ and month $m$:

$$SIC_{(i,q,m)} = \sum_{s=1}^{n_s} \left( w_{(i,s)} \times SIC_{opt_{(i,q,m,s)}} \right) \qquad (4)$$

with

$$w_{(i,s)} = \left( 1 + \left( \frac{d_{(i,s)}}{d_r} \right)^4 \right)^{-1} \qquad (5)$$

Here, $d_r$ is a reference distance of 500 km, yielding a smooth transition at the boundaries between two adjacent sectors. At the center of a sector, this yields a weight that is very close to 1 for the relevant field that was identified as optimal for that sector and that is close to 0 for the fields identified as optimal for the other sectors; at the boundary between two sectors, the weights are typically 0.5 for the two relevant sectors and close to 0 for the others.

### 2.4 Evaluation

Evaluation of the above methods is mainly achieved with a perfect model approach. In this test, we consider SST and SIC from the historical simulation of one coupled AOGCM as being the observations. Then, we apply the different bias correction methods using the climate change signal coming from a future scenario of the same AOGCM. Obtained future SST and SIC using this perfect model test are finally compared with original SST and SIC from the AOGCM climate change experiment. Additionally, we also performed an assessment of real case applications using observations and climate change signals coming from AOGCM projections. Changes in mean and variance in the coupled model projection with respect to the historical simulation are compared to the introduced change in mean and variance in the estimated future SST and SIC using bias correction methods with respect to the observed climatological data. We assume that an ideal bias correction method should reproduce the same change in mean and variance between the observations and the estimated future SST and SIC as between the used coupled GCM historical simulation and the climate change experiment.

## 3 Results

### 3.1 Sea Surface Temperatures

#### 3.1.1 Perfect model test

In this section, we discuss the application of the perfect model test for both the anomaly and the quantile-quantile method. To apply this test, we used CNRM-CM5 data from the historical simulation over the 1971-2000 period and from the rcp8.5 projection for the 2071-2100 period. Corrected rcp8.5 SST have been compared with the original SST projection. For the anomaly method, the relation between the anomaly-corrected future SST and the SST directly obtained from the AOGCM projection is trivial when we replace observed SST by SST from the AOGCM historical simulation in (1). As a result, when comparing corrected rcp SST using the perfect model test and original SST from CNRM-CM5 rcp8.5 scenario, we obtain,



**Table 1.** Mean and standard deviation difference between present and future SST data sets for North Atlantic (45°N to 58°N, 105°W to 85°W)

|  | Mean difference (°C) | STD difference (°C) |
|---|---|---|
| CNRM-CM5 rcp8.5 - CNRM-CM5 hist | +3.04 | +0.59 |
| Anomaly meth. app. - PCMDI obs | +3.06 | +0.66 |
| Quantile-quantile meth. app. - PCMDI obs | +3.04 | +0.68 |

by construction, a null bias all over the world (*figure not shown*). For the quantile-quantile method, the bias is also null in most regions. However, since we applied a very slight spatial filtering of the quantile increment, some slight biases (positive or negative) appear in regions of steep SST gradients (i.e. regions with major oceanic currents). Nevertheless, these biases are negligible (a few tenths of degrees Celsius; *figure not shown*).

### 3.1.2 Real-case application

Here, we present the application of the anomaly and the quantile-quantile methods in a real case application. For this application, we use SST data from PCMDI observations data set over 1971-2000, from the IPSL-CM5A-LR historical simulation over the same period, as well as the rcp8.5 scenario over 2071-2100. Histograms of frequency distribution of SST for different regions of the world (Weddell Sea, Central Pacific and North Atlantic) have been plotted in order to compare frequency distributions in the observations, in the GCM historical and future simulations, as well as in the estimated bias-corrected future SST using the quantile-quantile and the anomaly method (Figure 4). In this figure, we can appreciate the change in mean and variance between the GCM historical simulation and the GCM future scenario and between the PCMDI observations and in the estimated bias-corrected SST scenario. This change in mean and variance is more explicitly calculated and presented for the North Atlantic in Table 1. Results from the anomaly method and from the quantile-quantile method are very similar, and both methods succeed in applying the change in mean and variance coming from the AOGCM scenario to the observations.

### 3.2 Sea-Ice Concentration

### 3.2.1 Perfect model test

In this section, we present the results of the application of the perfect model test for the three methods for bias correction of SIC. The term "perfect model test" is not absolutely pertinent for the evaluation of the Look-up Table method, as we first computing look-up tables using SST and SIC from an AOGCM historical simulation. Then, we used the SST of the climate change projection from the same AOGCM and retrieved SIC with the help of the previously computed LUT. An example of computed LUT using data of the historical simulation of CNRM-CM5 can be seen in Figure 5. It is noteworthy that this new look-up table is significantly different from the one using PCMDI observations (Figure 2). Even though the use of this LUT for the perfect model test instead of LUTs computed using observed SST and SIC over the AMIP period can be discussed, the





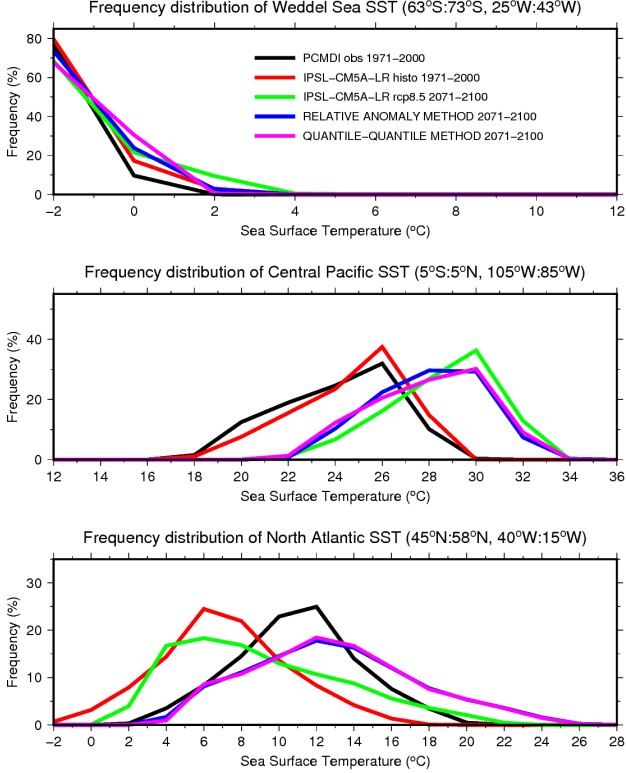

**Figure 4.** Frequency distribution of SST for PCMDI observations (black), IPSL-CM5A-LR historical (red) over 1971-2000 and rcp8.5 (green), quantile-quantile method (pink) and anomaly method (blue) applications over 2071-2100 for Weddell Sea (top), Central Pacific (center) and North Atlantic (bottom)

use of LUT computed using observations would necessarily produce poorer result for the reconstruction of SIC the AOGCM future scenario in a perfect model test. Using AOGCM data, inconsistent or missing results were found for most of SST bins at or below the freezing point of sea water (-1.8°C). In order to fill the LUT, we therefore fixed SIC=99% for SST=-2.0 °C and linearly interpolated SIC between -1.7°C and -2.0°C.

5    The perfect model test is more rigorously applied for the evaluation of the relative anomaly and the analog method, as we simply replaced time series of the observed SIC by the one from the AOGCM historical scenario before applying the method without any specific modification or calibration. For the analog method, we mention that the tested AOGCM projection has been excluded from the possible analog candidates before applying the method and the perfect model test.

Mean biases (%) after applying the perfect model test are shown for the three methods for the rcp4.5 and rcp8.5 scenarios

10   of the IPSL-CM5A-LR and CNRM-CM5 AOGCM (Figure 6). One can see that the mean bias on the estimation of sea-ice concentration remains reasonable for most of the Arctic and the Antarctic for the analog method and very small for the look-up table method. Some of the biases of the analog method for regions with very complex coastal geography, such as the Canadian





Archipelago, are due to the differences in land mask between the tested and the chosen AOGCM as analog candidate, despite the care taken for this issue. Mean bias for the relative anomaly method exhibits some larger values. The pattern of the biases using this method seems robust between the different AOGCM scenarios. It is also noteworthy that the pattern of the biases is also similar between different methods, especially if we consider the results in the Arctic for the scenarios of the CNRM-CM5

model.

With the results of the perfect model test, we also performed a comparison between the frequency distribution of the mean SIC in the AOGCM future scenario (here CNRM-CM5, rcp8.5) and in the corresponding estimation using the bias correction methods (Figure 7). In these plots, we represented the histogram of frequency of sea-ice concentration for four regions: Ross Sea (72°S:77°S; 174°E:163°W), Weddell Sea (63°S:73°S; 45°W:25°W), Arctic Basin (80°N:90°N; 180°W:180°E), and the

Canadian Archipelago (66°N:80°N; 130°W:80°W). These regions have been chosen because they are the principal regions where there remains a significant amount of sea-ice by the end of the 21$^{st}$ century under the rcp8.5 scenario. With the look-up table method (blue lines in Figure 7), the distribution of sea-ice concentration is more or less well reproduced in the Arctic (Figure 7 c and d), whereas in the Antarctic seas the distribution (Figure 7 a and b) exhibits well-marked peaks that we do not find in the GCM data set (black lines). The presence of such peaks is easy to explain by taking into account the structure of the

look-up tables. Moreover, using this method, we find a large underestimation of the sea-ice concentrations above 90%, mainly in the Southern Hemisphere, with almost no occurrence of these high sea-ice concentrations in the estimations using the LUT method for the Ross and Weddell Seas. The frequency distribution of the sea-ice using the relative anomaly method (green lines in Figure 7) seems more reasonable, even if there is a slight overestimation of the frequency for concentrations between 70 and 90% and an underestimation for very high sea-ice concentrations (above 90%). Finally the distribution obtained using

the analog method (red lines on Figure 7) is very close to the distribution of the original AOGCM future scenario. The results are robust because differences of sea-ice frequency distribution between future estimation and future AOGCM future scenario are very similar for rcp4.5 from CNRM-CM5 as well as for both scenarios from IPSL-CM5A-LR (*figures not shown*).

### 3.2.2 Real-case application

In this section, we applied the three bias correction methods using PCMDI SIC observations from the 1971-2000 period, as

well as the IPSL-CM5A-LR and CNRM-CM5 historical data over the same period and the data from the rcp4.5 and rcp8.5 future scenarios from 2071-2100 in order to obtain future bias-corrected sea-ice corrections. The reliability of the methods is evaluated by comparing the change in mean and variance between the observations over present climate and future estimated sea-ice concentrations to the corresponding changes in the climate change simulation with respect to the historical simulation. An ideal method should apply the same statistical changes to observed sea-ice as the one present in the climate change projec-

tion used to derive climate change signal.

In Figure 8, the bias-corrected mean sea-ice concentration change is plotted against the corresponding change in mean SIC in the AOGCM future scenario used to determine the climate change signal. All points in the plot are obtained by the same four AOGCM future scenarios as well as the same four "test regions" as in previous section (Ross and Weddell Seas, Arctic Basin, Canadian Archipelago). Similarly, in Figure 9, applied changes in standard deviation for the future estimated SIC are plotted





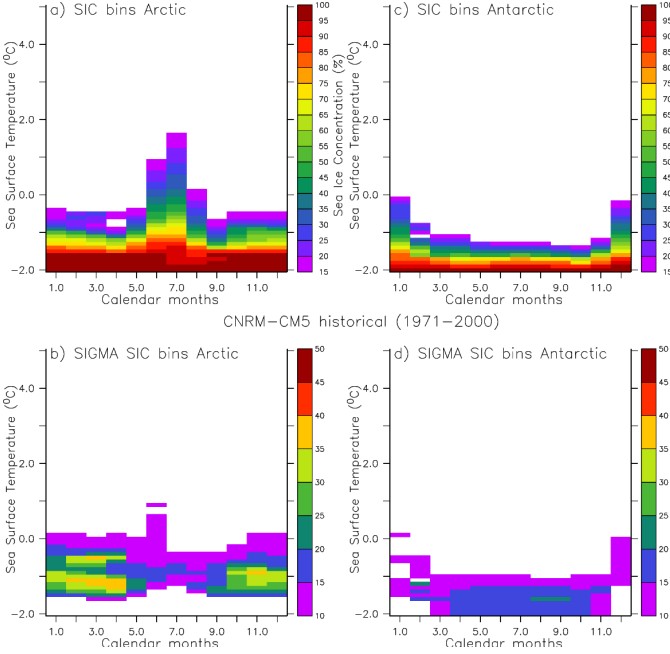

**Figure 5.** Look-up tables linking SST and SIC for the Arctic (a) and the Antarctic (c) built using 1971-2000 CNRM-CM5 historical simulation data and the associated uncertainty (root mean square error) on the computed SIC average (b,d)

against corresponding standard deviation change in the AOGCM climate change experiment.

For the look-up table method (Figure 8a), future SSTs have been bias-corrected using the quantile-quantile method before using computed LUT for the retrieval of future SIC. Using this method, there seem to be no systematic errors in the applied change in mean SIC. However, the spread of the points seems to increase for stronger decreases of sea-ice. Main outliers with

5 a high overestimation of the decrease in SIC are constituted by points representing the evolution of sea-ice in the Weddell Sea, mainly for CNRM-CM5 scenarios. If we consider change in SIC variability (Figure 9a), there is a strong systematic bias and the decrease in SIC variability in the future is strongly overestimated. Indeed, due to the structure of the look-up table itself, the variability of SIC in future estimations is much lower than in the observations.

The application of the relative anomaly method shows a more general overestimation of the decrease in mean SIC (Figure 8b).

This overestimation is more pronounced for the Weddell Sea area and for scenarios of the CNRM-CM5 model. Only the decrease in mean SIC in the Arctic Basin is correctly reproduced with respect to the AOGCMs future scenarios. Concerning the change in SIC variability (Figure 9b), the increase in variability in the Arctic Basin and in the Canadian Archipelago is correctly reproduced whereas for the Antarctic seas and particularly the Weddell sector, the decrease in SIC variability is once again massively overestimated.

Finally, the application of the analog method is able to reproduce a great part of the change in mean SIC (Figure 8c). Nevertheless, distinct outliers corresponding to the Weddell Sea sector are once again present for each AOGCM scenario, with a strong overestimation of the decrease in sea-ice. As for the relative anomaly method, the change in SIC variability (Figure 9c)





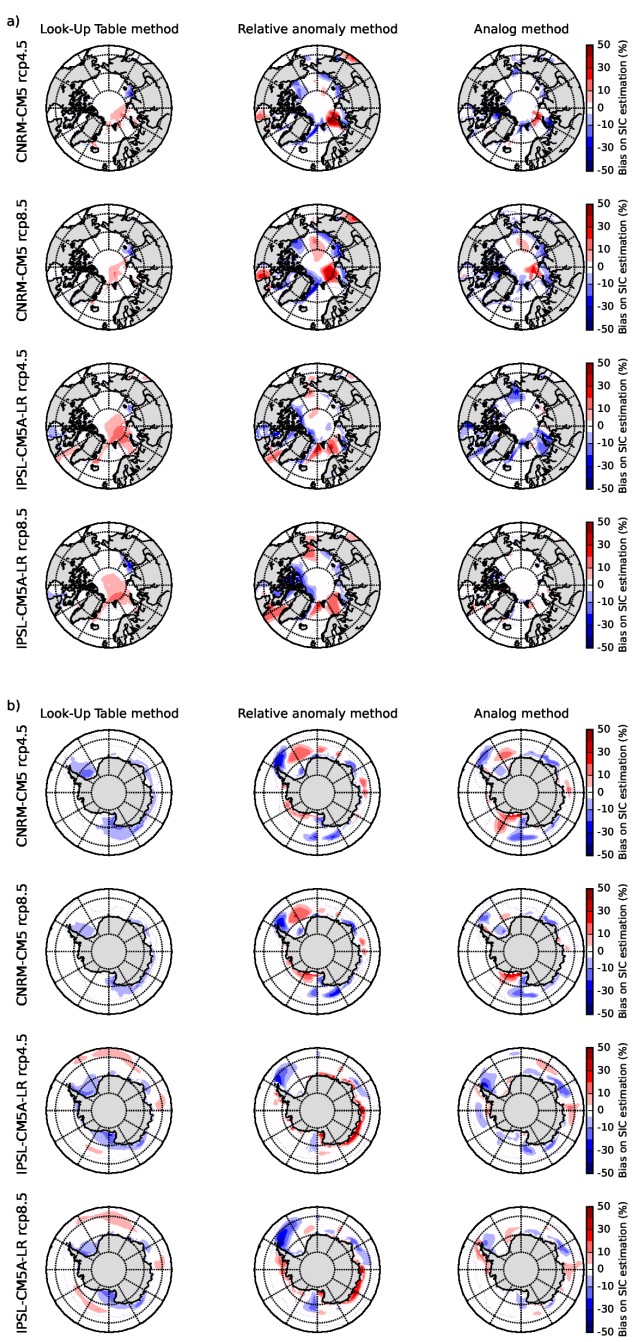

**Figure 6.** Mean bias on the estimation of SIC with respect to the original AOGCM future scenario for the LUT, iterative relative anomaly and analog method with CNRM-CM5 and IPSL-CM5A-LR rcp4.5 and rcp8.5 scenarios for the Arctic (a) and the Antarctic (b)





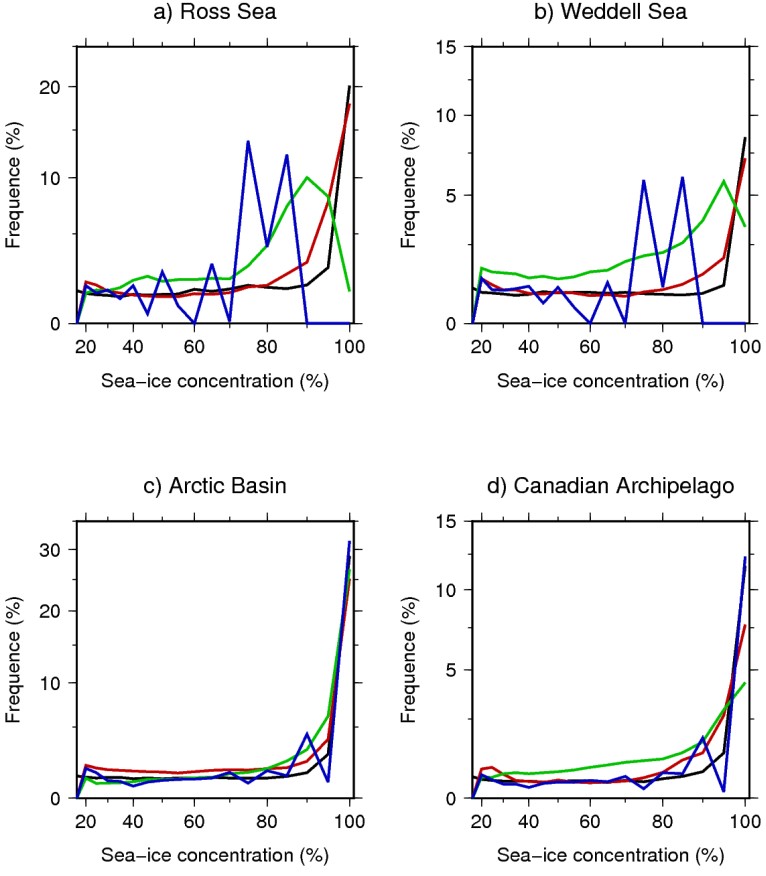

**Figure 7.** Frequency distribution of SIC in CNRM-CM5 rcp8.5 scenario (black) and in estimation using different methods in a perfect model test: Look-up table (blue), analog (red), and iterative relative anomaly (green). Regions are: a) Ross Sea (72°S:77°S, 174°E:163°W); b) Weddell Sea (63°S:73°S, 43°W:25°W); c) Arctic Basin (80°N:90°N, 180°W:180°E); d) Canadian Archipelago (66°N:88°N, 130°W:80°W)

is correctly reproduced in the Arctic, while there is a strong overestimation of the decrease in variability around Antarctica, particularly for the Weddell Sea.

### 3.3 Sea Surface Temperature and Sea-ice Concentration consistency

5  As bias correction of SST and sea-ice are performed separately, the physical consistency between the two variables is assessed a posteriori. To do so, three different issues are examined:

- There is a considerable amount of sea-ice (>15%) in the corrected scenario where the SST is above fresh water freezing point (273.15K). In this case, we set SST equal to the sea water freezing point (271.35K) for any SIC equal or greater





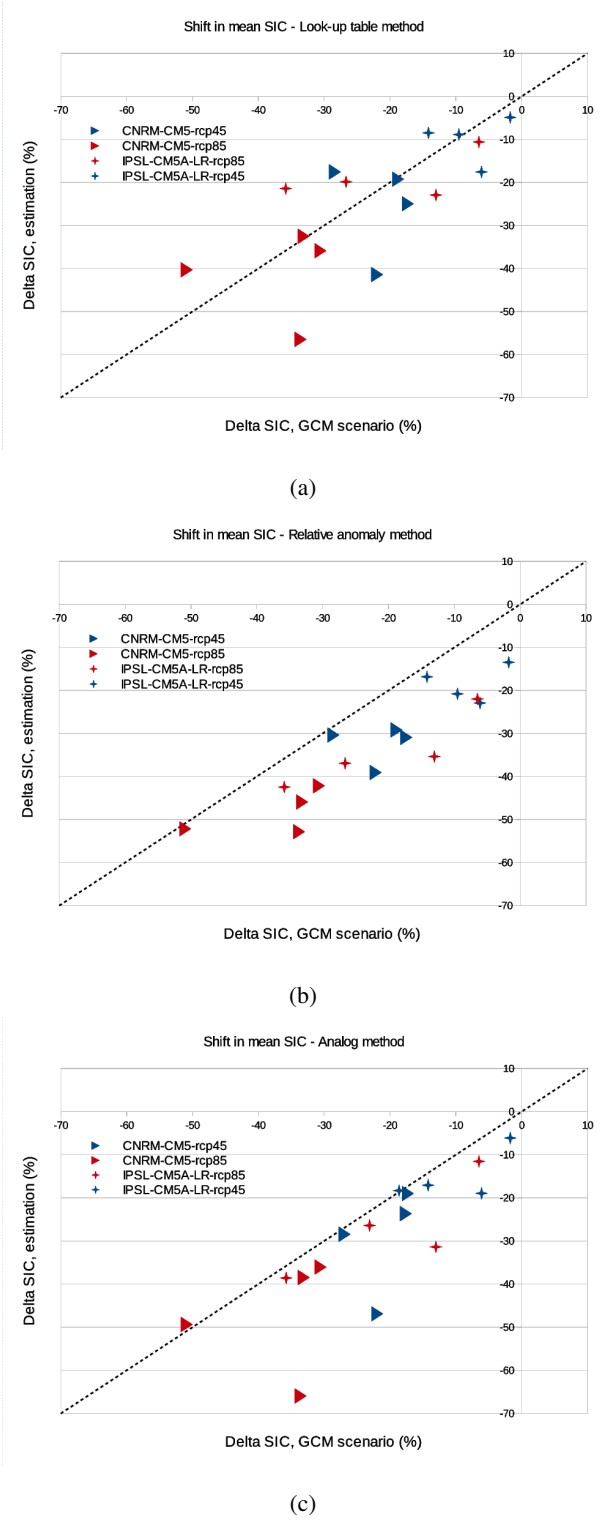

(a)

(b)

(c)

**Figure 8.** Change in mean estimated future SIC using a) look-up table, b) relative anomaly, c) analog method against corresponding mean change in the AOGCM future scenario for the four test regions (Canadian Archipelago, Arctic Basin, Weddell Sea and Ross Sea)





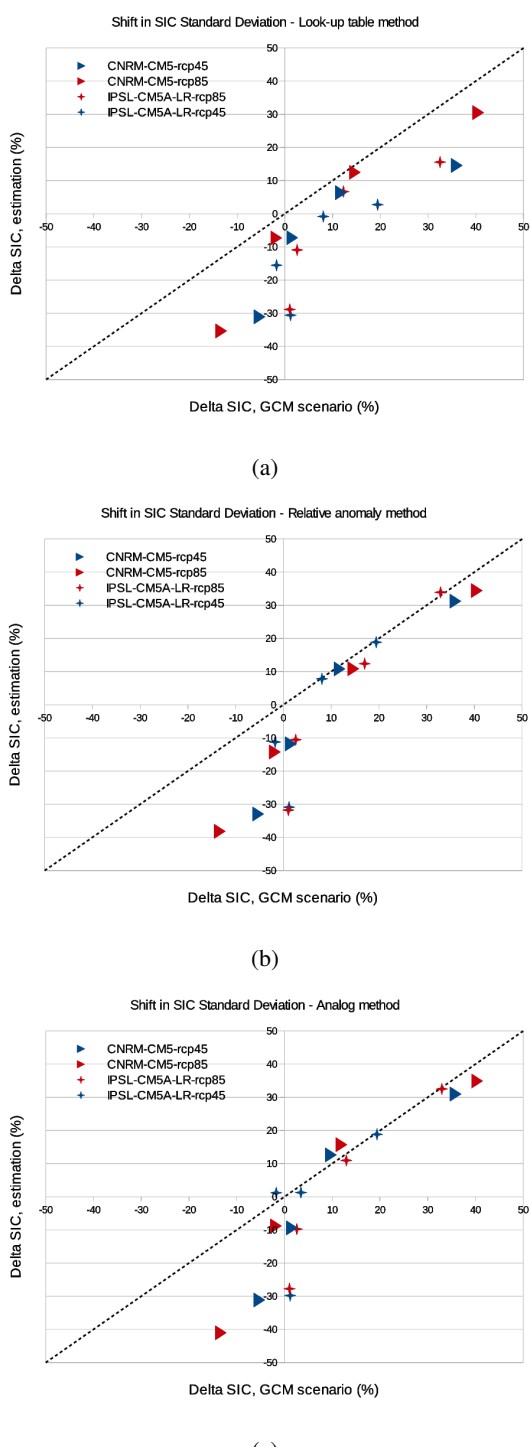

(a)

(b)

(c)

**Figure 9.** Change in estimated future SIC standard deviation using a) look-up table, b) relative anomaly, c) analog method against corresponding mean change in the AOGCM future scenario for the four test regions (Canadian Archipelago, Arctic Basin, Weddell Sea and Ross Sea)





than 50%. If the future calculated SIC is between 15 and 50%, the future SST is obtained by linearly interpolating between the sea water freezing point and the freshwater freezing point.

- The future corrected sea-surface temperature is below the fresh water freezing point but there is no significant (<15%) SIC in the bias-corrected scenario. In this case, we put the SST of the concerned grid point equal to the fresh water freezing point.

- SST has been used to remove very localized suspicious presence of sea-ice (no-ice) in the Arctic in summer. Any sea-ice for SST above 276.15K has been removed, this temperature being the highest temperature at which significant amount of sea-ice (15%) is found is the Arctic in the computed look-up table using PCMDI data.

The impact of these modifications has been evaluated using the framework of the perfect model test. After applying the analog method for SIC and the quantile-quantile method for SST in a perfect model approach, we applied the correction for SST and SIC consistency and compared obtained SSTs to the original AOCGM future scenario used to carry out the experiment. The biases can be seen in Figure 10 for the application of the method with IPSL-CM5A-LR and CNRM-CM5 scenarios. It is negligible in most regions. Very locally, it can reach up to 1°C. These regions generally correspond to regions where the analog method has shown some biases for the reconstruction of sea-ice especially for CNRM-CM5 scenarios. The occurrences of the three cases mentioned above have been assessed for both the perfect method test and the real-case application. First and third cases are very seldom and about 1% or less of the global oceanic surfaces experience at least one case during a 30 years experiment. The second case is more frequent, more than 20% of the global oceanic surfaces experience at least one occurrence during a 30 year experiment while the mean occurrence at each time step is about 1 to 2% of the global oceanic surfaces. This case is responsible for the small (0.25 to 0.5K) but widespread warm bias on SST that can be seen in the Antarctic seas for the reconstruction of IPSL model scenarios in Figure 10. Nevertheless, this slight decrease in quality of the reconstruction of SST is worth to consider in order to ensure physical consistency between SST and SIC.

## 4 Discussion

### 4.1 Sea Surface Temperatures

The bias correction of future sea-surface temperatures coming from AOGCM scenarios is an issue fairly easy to deal with and different appropriate solutions have already been proposed in the literature (e.g., Krinner et al., 2008; Ashfaq et al., 2011; Hernández-Díaz et al., 2017). In these papers, it has been demonstrated that the use of bias-corrected sea-surface temperatures has considerable influence on the modeled climate and its response in future scenarios for regions and processes as different as precipitation and temperature in the Tropics and the West African Monsoon as well as for the climate of Antarctica.

In this paper, we reviewed two existing bias-correction methods and propose a validation that allows objectively evaluating the efficiency of these methods with the use of a perfect model test and a real-case application. Since both methods show no bias in the perfect model test and succeed in reproducing the change in mean and variability coming from the AOGCM future scenarios, we can be confident in the use of these methods for bias-correction of future AOGCM scenarios.





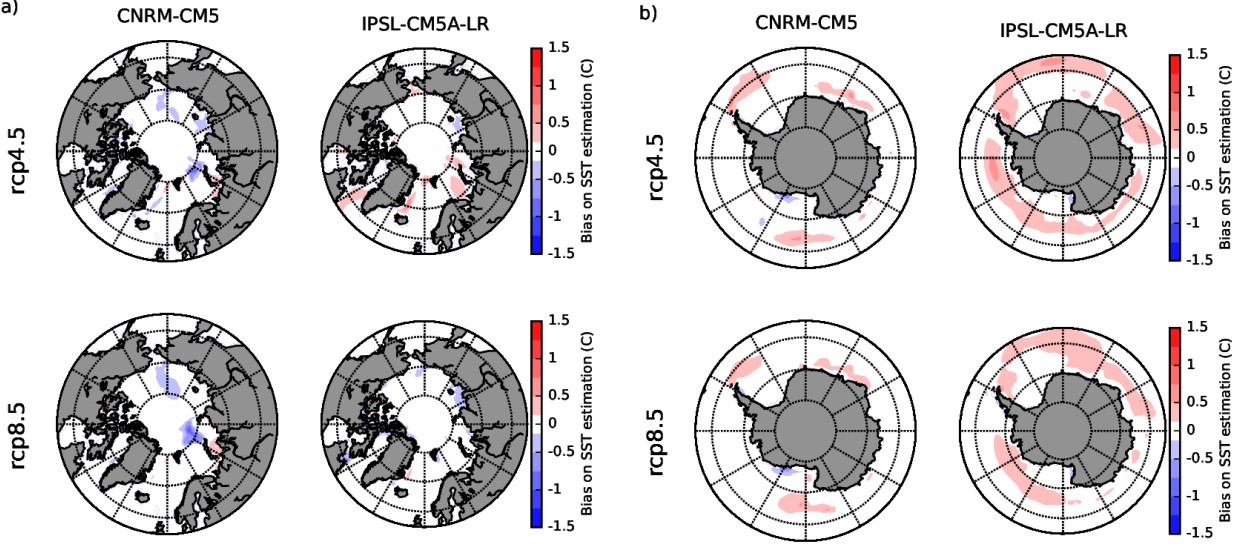

**Figure 10.** Mean bias on the estimation of the sea surface temperature with respect to the corresponding original AOGCM future scenario after applying the analog method for sea-ice, the quantile-quantile method for SST and the correction for SST and SIC consistency for the Arctic (a) and the Southern Oceans (b)

## 4.2    Sea-Ice Concentration

SIC is a quantity that has to remain strictly bounded between 0 and 100%, exhibits some sharp gradients and has to remain physically consistent with SST. Therefore the empirical bias correction of future SIC from coupled models scenarios is a much more complex issue to deal with than the bias correction of SSTs. The absence of satisfying solution proposals for this issue

5    in the literature has led to incorrect bias-correction of future SIC in a recent study (Hernández-Díaz et al., 2017). Yet, the proposal of convenient solutions for the bias correction of sea-ice for future scenarios is crucial for the community interested in the downscaling of future climate scenarios for polar regions.

In the perfect model test, we have seen that the look-up table method shows some reasonable mean bias over most regions (Figure 6). However, we have seen that the frequency distribution of future SIC obtained using this method is different from

10    the original distribution in the AOGCM and unavoidably exhibits some peaks due to the structure of LUT (Figure 7). Moreover, the absence of SIC above 90% in the Antarctic is also a considerable limitation to the method considering the large differences in terms of heat and moisture exchanges in winter between an ocean fully covered by sea-ice and an ocean that exhibits some ice-free channels (Krinner et al., 2010). In addition, the use of SST as a proxy for SIC is physically questionable, as we should expect a large SIC gradient around the freezing point. The fact that both SST and SIC are averaged over a long period (one

15    month) and over a considerable area (1°x1°) is probably the main reason why we find nevertheless a relation between the two variables. The real case application of the method also shows some difficulties for the reconstruction of large decreases in mean



SIC (Figure 8a) as well as a poor reconstruction of the change in variability in future SIC (Figure 9a).

The relative anomaly method (Krinner et al., 2008) shows the largest spatial mean biases in the perfect model test (Figure 6). The structure of some biases seems to be constant across the reconstruction of different climate scenarios used in the perfect model test. The empirical reduction of SIC by an iterative "erosion" from the edges of the sea-ice covered regions has most

likely the tendency to overestimate the decrease of sea-ice for some coastal regions, while it probably fails to reproduce some processes involved in the disappearance of sea-ice in the future such as for example the inflow of warmer waters through the Barents Sea or the Bering Strait in the Arctic. The "real-case" application of the relative anomaly method has shown some systematic negative bias in the reconstruction of the decrease in mean SIC (Figure 8b) and some important overestimation of the decrease in variability in the Antarctic seas (Figure 9b).

The evaluation of the analog method with the perfect model test allows to highlight some mean biases locally slightly bigger than for the look-up table method (Figure 6). However, the frequency distribution of the future estimated SIC perfectly reproduces the frequency distribution of the sea-ice in the original AOGCM future scenario (Figure 7). The real-case application of the method succeeds in reproducing the change in mean and variability of SIC for most of the tested regions and scenarios (Figure 8c). However, the decrease in mean (Figure 8c) and variability (Figure 9c) of the sea-ice in the Antarctic, particularly

the Weddell Sea, is also largely overestimated using this method. With respect to the relative anomaly method, the fact that we use observed or AOGCMs modeled sea-ice maps to reconstruct estimated future sea-ice and that we use a criterion for both sea-ice area and sea-ice extent allows us to better reproduce some critical features of future sea-ice and to obtain a more realistic frequency distribution. It should be noted that in the perfect model test as well as in the real-case application, the original AOGCM is not present among the possible analog candidates. If this is done, the results are even better using this method.

The fact that the analog method and the relative anomaly method share the same bias in the real-case application with a strong overestimation of the decrease in mean and variability of the sea-ice in the Weddell Sea in particularly for the scenarios of the CNRM-CM5 model is not a coincidence. For both methods, the targeted future sea-ice extent (area) for a given sector is a product of the division of the integrated sea-ice extent (area) in the AOGCM future scenario by the corresponding quantity in the historical simulation. As a consequence, the targeted future sea-ice extent (area) for a given sector and a given month is

null when the integrated sea-ice extent (area) is null in the future AOGCM scenario. Therefore, the bias in the future scenario is not corrected in that case. The fact that both methods overestimate the decrease in sea ice mainly for CNRM-CM5 scenarios is to be linked to the fact that the historical simulation of this AOGCM shows some considerable negative biases for the sea-ice in the Weddell Sea with respect to the observations. Consequently, SIC in the Weddell Sea in CNRM-CM5 future scenario is low and the number of months with a complete disappearance of sea ice is large. For these months, SIC in these sectors is not

bias-corrected with the latter two methods. This means that although the methods described here are in principle applicable to any AOGCM output, it seems to be wise to preferentially select output of reasonably "well-behaving" AOGCMs as initial material for the bias-correction.



### 4.3 A note on sea-ice thickness

Air-sea fluxes in the presence of sea ice are strongly influenced by the thickness of the sea ice and the overlying snow cover. Gerdes (2006) and Krinner et al. (2010) have shown that the atmospheric response to changes in Arctic sea-ice thickness is substantial. In most AGCMs, sea-ice thickness will also need to be prescribed along with sea-surface temperature and sea-ice

concentration. When SST and SIC from a coupled climate model are directly used, sea-ice thickness from that same run should of course be used; however, in case SST and SIC from the coupled model run are bias-corrected, as we strongly suggest here, we argue that sea-ice thickness should be prescribed in a physically consistent manner in the atmosphere-only simulation. An in-detail evaluation of sea-ice thickness prescription methods is beyond the scope of the main part of this paper. Therefore, an evaluation and further refinement of a simple parameterization of sea-ice thickness as a function of instantaneous and annual

minimum SIC, initially suggested by Krinner et al. (1997) and used by Krinner et al. (2010), is presented in the supplementary material of this paper.

### 4.4 General considerations on bias correction of oceanic forcings

As already mentioned before, one may doubt whether it is possible to bias-correct a GCM that has overly strong biases in present-day climate. Indeed, most of the bias-correction methods rely on the hypothesis than the climate change signal coming

of an AOGCM scenario is not dependent on the bias in the historical simulations. This hypothesis can largely be questioned in a non-linear system (formed by SIC and SST). For example, in a model with a strong negative bias in sea-ice for present-day climate most of the additional energy due to an enhanced greenhouse effect will be used to heat the ocean while it would be primarily used to melt sea-ice in a model with a correct initial sea-ice state. For such a model, the reliability of the climate change signal in SST is thus necessarily questionable.

Skills of coupled GCMs in reproducing the observed climate and its variability for a region of interest are often evaluated in order to use the GCM output as forcing for downscaling experiments. However, skills of atmospheric GCM are generally better when forced by observed oceanic boundary conditions (Krinner et al., 2008; Ashfaq et al., 2011; Hernández-Díaz et al., 2017). Similarly, even though bias correction methods have some limitations, for future climate experiments, there are good reasons to believe that simulations produced using bias-corrected oceanic forcings bear reduced uncertainties with respect to simulations

realized with "raw" oceanic forcings from coupled model scenarios such as those from the CMIP5 experiments.
Bias-corrected oceanic forcings can be used to force a regional climate model (RCM), but in this case an additional modelling step has to be carried out, as bias-corrected oceanic forcings should be used to force an atmosphere only GCM that will provide atmospheric lateral boundary conditions for the RCM in order to ensure the consistency between oceanic and atmospheric forcings, such as in Hernández-Díaz et al. (2017) In this framework, the use of a variable resolution GCM which allows to

directly use bias-corrected oceanic forcings and downscale future climate experiments is an alternative worth considering, as it also allows two-way interactions between the downscaled regions and the general atmospheric circulation.





## 5 Conclusions

In this paper, we reviewed existing methods for bias correction of SST and SIC and proposed new ones, such as the analog method for sea-ice. We also proposed validation methods that allow objectively evaluating bias-correction methods with the use of a perfect model test and real-case applications.

The bias-correction of SST is an issue that has already been widely addressed in recent papers and its importance for the modeling and downscaling of future climate scenarios has been demonstrated for multiple regions of the world. In our analysis, we were able to demonstrate the reliability and the suitability of absolute anomaly and quantile-quantile methods for the bias correction of future SST scenarios.

The bias correction of SIC is a more difficult issue to address. With the analog method, we propose a method that shows
promising results in most cases and that allows reconstructing future SIC with a realistic frequency distribution in the future. However, the fact that the relative anomaly between an AOGCM future scenario and the historical simulation is also used in this method in order to determine future targeted sea-ice extent and area, prevent from bias-correcting cases where sea-ice disappears entirely in a given sector or even an hemisphere. Despite the absence of a perfect and definite answer to this issue, we propose a new and improved method as well as a convenient, objective way to evaluate bias correction methods for
future climate scenarios. We draw the attention on the bias-correction of sea-ice that is currently somewhat overlooked by the community. The application of a multivariate bias correction method (Cannon, 2016) is also a perspective that could help with the bias correction of SST and SIC future scenarios at the same time. Nevertheless, corrected SIC using the analog method represent a substantial improvement with respect to other previously existing bias-correction methods for sea-ice scenarios and will therefore be made available to anyone willing to use them as forcing for bias-corrected downscaling experiments.

*Code and data availability. FORTRAN* code enabling the generation of bias-corrected future SST and SIC using CMIP5 scenarios and PCMDI data as input are publicly available for each method via *https* transfer (https://mycore.core-cloud.net/index.php/s/3Lo3Tlr9wsyUGjk) or *ftp* transfer (ftp://ftp.lthe.fr/pub/beaumet/Sourcecode_SSTSICmethods.tar.gz). Bias-corrected future CMIP5 scenarios (rcp4.5 and 8.5) realized within the frame of this study (IPSL-CM5A-LR and CNRM-CM5) are available as well (https://mycore.core-cloud.net/index.php/s/Q1cIsS71Mo4vC or ftp://ftp.lthe.fr/pub/beaumet/Data_BCSST-SIC.tar.gz).

## Appendix A: A simple diagnostic parameterization of sea-ice thickness for AGCM simulations

### A1 Introduction - general remarks

Atmospheric circulation models (AGCMs or regional climate models) require information about the state of the sea surface as a lower boundary condition. While much attention has been paid to sea-surface temperature (SST) and sea-ice concentration (SIC) in that respect, the issue of prescribing correct (or at least reasonable) sea-ice thickness (SIT) has been somewhat
neglected historically. While there is a considerable body of scientific literature on the effect of varying SST and SIC on





simulated climate, only very few studies focused on the role of varying SIT in atmosphere-only simulations. The authors are aware of three such studies (Gerdes, 2006; Krinner et al., 2010; Semmler et al., 2016). Gerdes (2006) concluded that "realistic sea ice thickness changes can induce atmospheric signals that are of similar magnitude as those due to changes in sea ice cover", while Krinner et al. (2010) show that the impact of a variable sea-ice thickness compared to a uniform value is

essentially limited to the cold seasons and the lower troposphere, and that sea-ice thickness changes have a significant impact also in the context of climate change simulations. Near-surface temperature changes of the order of a few °C are observed in response to the replacement of a uniform thick Arctic sea-ice cover by variable sea-ice thickness.

In this note, a simple diagnostic parameterization initially developed by Krinner et al. (1997) is discussed and evaluated against new Arctic and Antarctic sea-ice thickness data that were not available in the mid-90s.

The idea is to propose a simple parameterization of sea-ice thickness that can be used in a variety of climate modelling applications, in particular for AGCM or RCM simulations of climate conditions different than today, from palaeoclimate studies to climate projections. In these applications, this parameterization can be particularly useful in cases where future sea-surface conditions (SST, SIC and SIT) are not directly prescribed from a coupled ESM run, but rather obtained using a bias correction method.

## A2    Methods

### A2.1    Diagnosing sea-ice thickness from sea-ice concentration

As described by Krinner et al. (2010), the parameterization of sea-ice thickness $h_S$ as a function of the local instantaneous sea-ice fraction f is designed such as to yield $h_S$ of the order of 3 meters for multi-year sea ice (deemed to be dominant when the local annual minimum fraction $f_{min} \gg 0$) and $h_S$ below 60cm (with a stronger annual cycle) in regions where sea-ice

completely disappears in summer (that is, $f_{min} = 0$), and intermediate values for intermediate cases:

$$h_S = (c_1 + c_2 f_{min}^2) \cdot (1 + c_3(f - f_{min})) \tag{A1}$$

with $c_1$=0.2m, $c_2$=2.8m and $c_3$=2. This corresponds to the observed characteristics of Arctic and Antarctic sea ice, with multi-year sea ice being generally much thicker than first year ice. The parameter $c_3$ introduces a seasonal ice thickness variation in areas where there is a concomitant seasonal cycle of sea-ice concentration. A more parsimonious, simply bilinear formulation

could have been designed to comply with these constraints. However, for the sake of consistency with previous work, we used the equation proposed by Krinner et al. (1997) who designed the parameterization such as to allow for a fairly strong seasonal cycle of sea-ice thickness also in regions with intermediate values of $f_{min}$. Figure A1 (from Krinner et al. (2010)) illustrates diagnosed Arctic sea-ice thickness for the present and for the end of the 21$^{st}$ century (2081-2100) using bias-correction applied to sea-ice concentrations from a coupled ESM SRES-A1B simulation (Krinner et al., 2008).





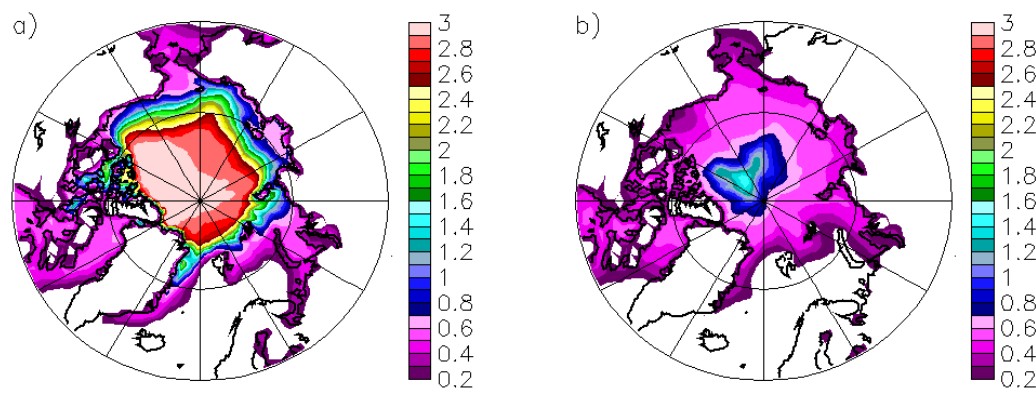

**Figure A1.** Prescribed annual mean Arctic sea-ice thickness (in m) in an AGCM climate change experience with bias-corrected sea-surface conditions, using the proposed diagnostic parameterization (figure from Krinner et al. (2010)). Left: present (1981-2000), right: SRES-A1B for 2081-2100. Bias correction for SST and SIC after Krinner et al. (2008)

### A2.2   Data

In the following, we used sea-ice concentration data extracted from the ERA-Interim output; this is typically the kind of data that would be used in AGCM or RCM simulations.

Lindsay and Schweiger (2015) recently proposed a 15-parameter spatial and temporal regression of Arctic sea-ice thickness observations from submarines, aircraft and satellites. We will use these observations here. Kurtz and Markus (2012) have deduced Antarctic SIT from ICESat data for the period 2003-2008. Although observations with autonomous underwater vehicles by Williams et al. (2015) tend to suggest occurrence of thicker Antarctic sea-ice than previously acknowledged, we will use the Kurtz and Markus (2012) data because of their large spatial coverage.

### A3   Results

The original formulation by Krinner et al. (1997) was parameterized for both hemispheres. We will therefore first present results for the original unique parameter set $c_{1,2,3}$ applied to both hemispheres. In a second step, we will present results for separate Arctic and Antarctic parameter sets, yielding a better fit to the observations. The reasoning is that, at the expense of generality of the diagnostic parameterization, one could argue that the strong difference between the Arctic and Antarctic geographic configuration — a closed small ocean favouring ice ridging and thus thicker sea ice in the Arctic, and large open ocean favouring thinner sea ice around Antarctica — justifies choosing different parameter sets for the two hemispheres. As





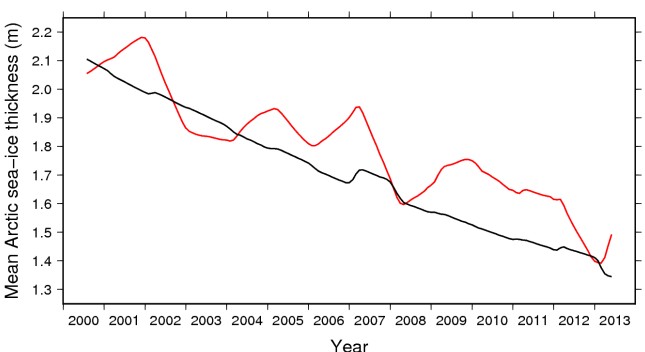

**Figure A2.** Observed (black, after Lindsay and Schweiger, 2015) and diagnosed (red) 12-month moving average mean sea-ice thickness of the Arctic basin (see Figure A3). The global parameter set is used here. Slight differences to Figure A4 of Lindsay and Schweiger (2015) appear because here we mask ice-free (SIC < 15%) areas that have a finite, non-zero ice thickness in the regression proposed by Lindsay and Schweiger (2015) who extend their regression to the entire Arctic Basin at all seasons.

the position of the continents will not change over the time scales of interest here, climate change experiments will not be adversely affected by this loss of generality.

### A3.1 Option 1: Global parameter set

A comparison between the observed (Lindsay and Schweiger, 2015) and our diagnosed evolution of the Arctic mean sea-ice
thickness is given in Figure A2. The geographical patterns of the observed (in fact, observation-regressed) and parameterized Arctic ice thickness for March and September over the observation period 2000-2013 (Figure A3) do bear some resemblance, but they also show some clear deficiencies of the diagnostic parameterization. The diagnostic parameterization reproduces high sea-ice thickness north of Greenland and the Canadian Archipelago, linked to persistent strong ice cover, but underestimates maximum ice thickness (due in part to compression caused by the ocean surface current configuration). Thinner sea ice over
the seasonally ice-free parts of the basin is reproduced, but it is actually too thin, particularly in winter (for example in the Chukchi Sea). Obvious artifacts appear in September north of about 82°N where the SIC in the ERA-Interim data set clearly bears the signs of limitations due to the absence of satellite data.

Both for spring (Oct-Nov) and fall (May-Jun), our diagnosed SIT (Figure A4) compares generally well with the ICESat data except for an overestimate in the Weddell Sea, at both seasons. The geographical pattern of alternating regions with thin and
thick sea ice is remarkably well reproduced.

### A3.2 Option 2: Separate Arctic and Antarctic parameter sets

A slightly better fit for the two poles can be obtained with separate parameters sets. For the Arctic, it seems desirable to increase winter sea-ice thickness in the Chukchi Sea area (by increasing $c_3$ slightly) and to decrease the average sea-ice thickness over





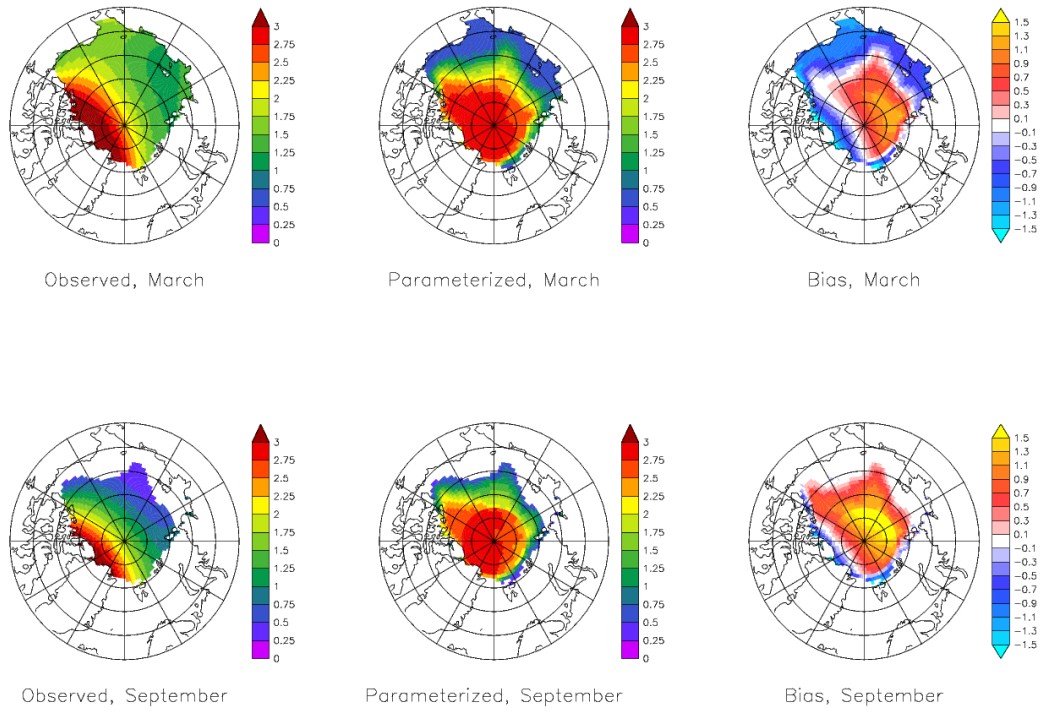

**Figure A3.** Observed (regressed, Lindsay and Schweiger (2015)) and parameterized Arctic sea-ice thickness (in m) for March and September, and difference between these (right), with the global parameter set.

the Central Arctic (by decreasing $c_2$). Figures A5 and A6 show results for the Arctic with $c_1$=0.2m, $c_2$=2.4m and $c_3$=3. The spatial fit is slightly better, but the recent Arctic-mean decadal tendency towards decreased average sea-ice thickness is somewhat less well reproduced. For the Antarctic, the main feature to improve is the maximum ice thickness in the Weddell Sea, which can be decreased by decreasing $c_2$ to 2.0m. The Antarctic parameter set then becomes $c_1$=0.2m, $c_2$=2m and $c_3$=2.

5 The result (Figure A7) is indeed a decreased thickness of the perennial Weddell Sea ice with little impact elsewhere.

In any case, these hemisphere-specific sea-ice parameter sets are not very different from each other and fairly similar to the original formulation.

## A4 Discussion and conclusion

Given the simplicity of the proposed diagnostic sea-ice thickness parameterization, the results are, at least in some aspects

10 such as the predicted average Arctic sea-ice thinning, surprisingly good. The Central Arctic sea-ice thickness results are clearly adversely affected by the input sea-ice concentrations north of 82°N. Arctic winter sea-ice thickness in the marginal seas appears underestimated. In the Antarctic, the spatial pattern of SIT is very well represented.





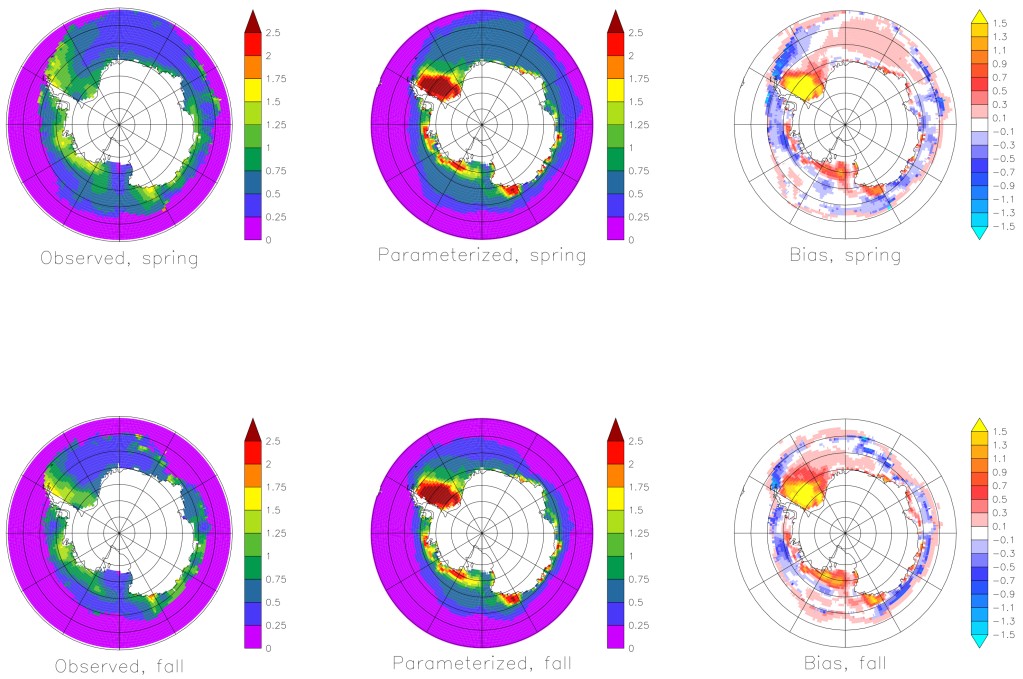

**Figure A4.** Observed Kurtz and Markus (2012) and parameterized Antarctic sea-ice thickness (in m) for March and September, and difference between these (right), with the global parameter set.

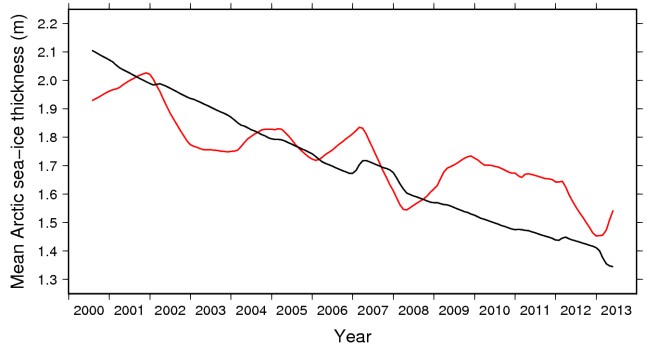

**Figure A5.** Observed (black, after Lindsay and Schweiger (2015)) and diagnosed (red) 12-month moving average mean sea-ice thickness of the Arctic basin with the Arctic-specific parameter set.





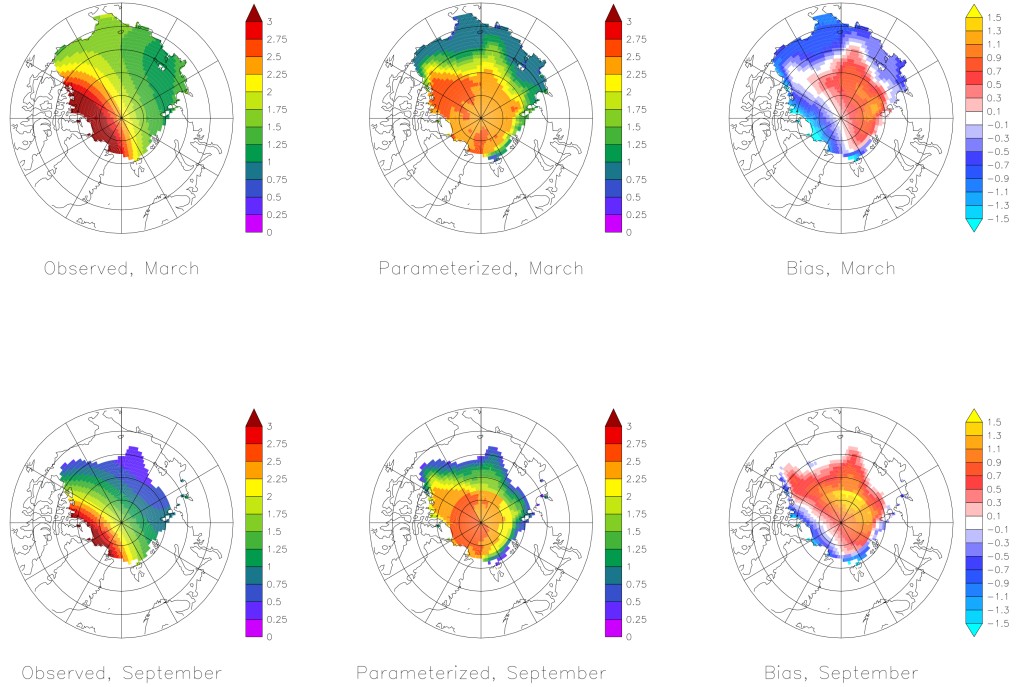

**Figure A6.** Observed Lindsay and Schweiger (2015) and parameterized Arctic sea-ice thickness (in m) for March and September, and difference between these (right), with the Arctic-specific parameter set.

We think that in absence of pan-Arctic and pan-Antarctic satellite-based data before approximately 2000, this parameterization can serve as a surrogate for earlier periods, and that it can, because it seems to have predictive power, also serve for climate change experiments with AGCMs or RCMs. Because of its simplicity, implementing this parameterization should not be too complicated in any case provided the model does explicitly take into account sea-ice thickness in its computations of heat flow through sea ice. In that case, sea-ice thickness can either be calculated online (with the need to keep track of annual minimum sea-ice thickness during the execution of the code) or be input as a daily boundary condition along with the sea-ice concentrations.

Of course, another possibility would be to prescribe sea-ice thickness anomalies from coupled models. In this case, it would probably be wise to compute the prescribe SIT using relative sea-ice thickness changes. For example, in a climate change experiment, this would read $h_{presc}(t) = h_{obs, 2003-2008} \cdot h_{sim}(t)/h_{sim, 2003-2008}$.

In any case, it is very probable that Arctic sea ice thickness will further decrease as multi-year sea ice will be replaced by a predominantly seasonal sea-ice cover. This should probably be taken into account in future CORDEX- or HighResMip-style climate simulations, given the non-negligible impact of sea-ice thinning on winter heat fluxes in particular.



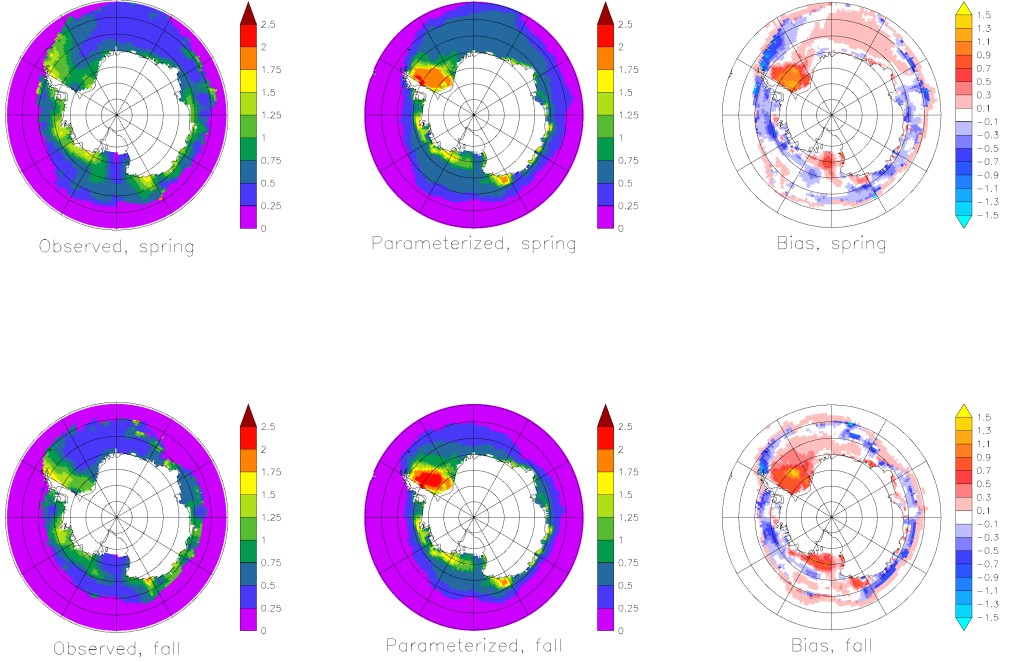

**Figure A7.** Observed Kurtz and Markus (2012) and parameterized Antarctic sea-ice thickness (in m) for March and September, and difference between these (right), with the Antarctic-specific parameter set.

*Competing interests.* This article has no competing interests.

*Acknowledgements.* This study was funded by the Agence Nationale de la Recherche through contract ANR-14-CE01-0001-01 (ASUMA). We acknowledge the World Climate Research Programme's Working Group on Coupled Modelling, which is responsible for CMIP, and we thank the climate modeling groups participating to CMIP5 for producing and making available their model output. For CMIP the U.S.

5   Department of Energy's Program for Climate Model Diagnosis and Intercomparison provides coordinating support and led development of software infrastructure in partnership with the Global Organization for Earth System Science Portals.

R. Haarsma was funded through the PRIMAVERA project under Grant Agreement 641727 in the European Commission's Horizon 2020 research program.

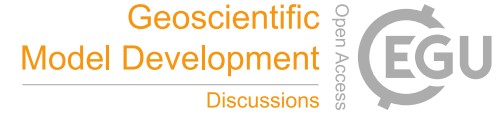

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
