# Peer review of "Assessing bias-corrections of oceanic surface conditions for atmospheric models"

_Geoscientific Model Development, 2017_

## Short Comment (SC1) · 22 Dec 2017

As explained in https://www.geoscientific-model-development.net/about/manuscript_types.html GMD is promoting that the program code (including data) described in the manuscript is publicly available through a permanent arrangement. For your manuscript we would like to encourage you to provide data and program code as supplemental material as this would be the easiest mechanism to achieve this; in fact some of the links provided in the manuscript did not work.

Lutz Gross GMD Executive Editor

---

## Short Comment (SC2) · 22 Dec 2017

Dear Editor, dear all,

I apologize for the current difficulties to access data and source code. Some links provided in the "code and data availability" section of the paper do not work. Indeed, the ftp server of my laboratory currently undergoes some technical issues. The https links are still working. Unfortunately, unless I am mistaking, It does not seem possible for me to upload them as supplementary material as long as the paper is in review process? As a temporary solution, I propose to upload these files as supplement of this comment.

Regards, Julien Beaumet

[Figure]

Please also note the supplement to this comment:
https://www.geosci-model-dev-discuss.net/gmd-2017-247/gmd-2017-247-SC2-
supplement.zip

──────────────────────────────

---

## Short Comment (SC3) · 22 Dec 2017

Dear all,

The zip files attached to the previous comment is empty. My apologies for this error. Please find a proper zip file attached to this comment.

Regards, Julien Beaumet.

Please also note the supplement to this comment: https://www.geosci-model-dev-discuss.net/gmd-2017-247/gmd-2017-247-SC3-supplement.zip

---

## Referee Comment (RC1) · F. Gallo (Referee) · 10 Jan 2018

General comments This paper proposes a way to evaluate bias-correction methods for SST and SIC for future climate projections, using a perfect model approach and a real-case application. There has clearly been a large amount of work in this study and this is clear when reading the paper. The analysis is thorough and the discussion honest, with the main caveats being highlighted and explained (at least, an explanation is proposed). The conclusion is clear and includes potential other methods to investigate. However, the presentation of methods and results might be a little bit confusing, given the amount of data. Some extra introductory sentences explaining the point of using a perfect model approach would be welcome, as it might not be obvious to a reader that is not a specialist but wants to learn more. This could be done in section 2.4, where the

description of the evaluation method (which is a main point of the paper) is a bit short. More generally, it would be interesting to provide some examples of the use of a perfect model approach in the literature. If the language is usually clear and understandable, the wording can be unusual and the authors are encouraged to have (another?) correction by a native speaker (which the reviewer is not...). On a more specific point, onw might wonder why were the GCMs CNRM-CM5 and IPSL-CM5A-LR chosen? Was this a choice based on availability or were these models selected based on their respective performance for representing SST and SIC? It would be nice to have information on this point. Finally some caveats and issues are treated too lightly and would require a more thorough description and explanation (see specific comments). Overall, the proposed paper describes an interesting and detailed work that should be of interest to many users in the climate modelling community. I therefore propose this manuscript to be accepted after the minor changes described in this review document.

Specific comments P1 L12: the part about RCPs is not needed, isn't it? The sentence is a bit long L17 bracket missing somewhere L19: Would it be possible to have some other examples from the literature? Surely a list of 4 or 5 references should be easy to find L19 "For example, it has..." L20 the seasonal cycle and the trend P2 L24 describe "AMIP" as "Atmospheric Model Intercomparison Project" if it is not done anywhere else P3 L25 Is there a reference for the Hann box filter? Why did you choose this filter? P5 L13 Any information on the number / proportion of GCMs that were dismissed? What does "poorly" mean for the selection process? L13 AOGCMs and remove"overly" P7 L12 "We assume that an ideal bias correction method should reproduce the same change in mean and variance between the observations and the estimated future SST and SIC as between the used coupled GCM historical simulation and the climate change experiment." That seems obvious but is there any reference regarding this issue? Is there any discussion among the scientific community? L23 What is the point of applying the perfect model approach for SST, as we use only "regular" bias correction? You highlight this issue, but you might want to shrink this section a bit. P8 Fig4 Are you sure about the color? There seems to be a very large initial

bias between the obs and the historical simulation for North Atlantic, is that expected? Moreover the RCP4.5 looks quite cold compared to the corrected values. If this is correct, can you highlight and explain that in the text? L11 "methods" L12 delete "in" P9 L9 This comment is valid for the whole paper, but is the use of "biases" valid when describing the results of the perfect model experiment? It is a bit confusing with the original bias that we are trying to correct. Again, if it has been used previously in the literature in that context, I'm ok, but maybe "difference" or "error" would be clearer, as it is a bias created by the method, and not a bias originally in the data P10 L12 "more or less" – can we find a more scientific term please? L14 "is easy to explain" – Is it? Cand you develop, please? L29 Should an ideal method apply the same statistical changes? It sounds right, but what about skewed distribution (precipitation) where the BC would change the distribution, therefore changing the distribution of changes? I think there is quite a discussion about that topic, so, if I agree with you, I would change to "We consider here that an ideal method. . ." P11 All text – Would it be possible to have some correlation value in order to quantify the error among the different methods? Maybe a correlation coefficient, or the value of the minimum, maximum and mean error for each graph? Fig8 and 9 It is difficult to see which point correspond to what – Maybe adding a letter to each of them to point to the region would help – Please try but it might make the figure impossible to read. It would be nice to be able to navigate alone within the points

---

## Referee Comment (RC2) · Anonymous Referee #2 · 2 Feb 2018

This manuscript discusses bias correction of sea-surface temperature using the anomaly method and the quantile-quantile method, and bias correction of sea ice concentration using the look-up table method, the iterative relative anomaly method, and the analog method. These bias correction methodologies are evaluated using a perfect model test (i.e. evaluated using the given model as "observations") and a real-case application in which the bias correction methods are compared to observations. It is assumed that ideal bias correction will reproduce changes in the mean and variance between observations and projected climate as between historical simulations and projected climate. The authors determine that the presented methods for bias correcting SST are reliable. The methods presented for sea-ice concentration are less reliable, however, the analog method showed promising results and improvement over

other bias correction methods. Additionally, the authors provide an appendix with a proposed method to parameterize sea-ice thickness, with potential for use in climate modeling applications.

I have a number of major and minor comments for the authors to address. Some of the manuscript was unclearly written, making the arguments difficult to follow. I also question the inclusion of SST bias correction evaluation. In regards to the review criteria, the manuscript does present relevant information that is related to modeling questions, particularly for sea-ice concentration, rendering it suitable for publication. However, much of the methodology for sea-surface temperature bias correction has been noted in other manuscripts.

My comments are below.

General Comments: 1. While the presented results for SIC are novel and will be very helpful for future modeling studies, the presented results for SST are somewhat less of an advancement. SST bias correction has been studied previously. In fact, there is much less discussion surrounding SST bias correction, and the results are almost glossed over by the authors in comparison. While the results are helpful in a summary sense for an interested reader, the concept seems less novel. This section may be able to be reduced even more, or eliminated completely. 2. The Appendix describes a methodology for parameterizing sea-ice thickness, which was noted in Section 4.3 as a strong influence. While you state that an in-detail evaluation of sea ice thickness prescription is beyond the scope of this paper, you evaluate and further refine one of the methods for parameterization in the Appendix. This seems like an important contribution to the field that has been studied comparatively less than, for example, SST bias correction methodologies. I'm concerned that this contribution will be lost due to its presence in supplementary material, and would potentially warrant a separate manuscript that delves more deeply into the topic. 3. I am curious why the CNRM-CM5, IPSL-CM5A-LR, and HadGEM-ES coupled GCM data were explicitly chosen for this study. In addition, you note that HadGEM-ES was used in Section 2.1 near line 25, but

never mention results from this model. 4. I am also curious why you selected the given bias correction methods for SST and SIC, are these arguably the most popular methods in use? If so, it would be helpful to note this as a motivation for the work. 5. Because you are using a perfect model test, can these results be generalized to other models, or are these results specific to the models used? 6. The introduction could benefit from additional discussion on SST biases, as it is written the focus is on SIC biases. 7. Figure 6 and resulting discussion: How does one determine what is a "reasonable" and "very small" error? To me, these look like large errors overall, but perhaps they are reasonable and very small with respect to the relative anomaly method? 8. In Section 4.2, page 18, last sentence on the page: Preferentially selecting output of reasonably "well behaving" AOGCMs is perhaps too simplistically stated here. There are a variety of issues in selecting which models are "well behaving". Though the following reference focuses on selecting models for regional hydrological studies, some of the general comments will still hold true for model selection: Brekke LD, Dettinger MD, Maurer EP, Anderson M (2008) Significance of model credibility in estimating climate projection distributions for regional hydroclimatological risk assessments. Clim Change 89:371–394 . doi: 10.1007/s10584-007-9388-3 9. Is the main result for SST bias correction that either method is appropriate for use due to your evaluation of the reliability of these methods? How does this result differ from other work on SST bias correction?

Technical Comments: The following comments should be easy to address, but will substantially improve the readability of the manuscript. 1. Please confirm that all acronyms are clearly defined, I have not listed all instances, but a few examples follow: CMIP5, AOGCM, AMIP, PCDMI, AGCM, etc. 2. Please confirm that all acronyms are consistent throughout the manuscript. I have not listed all instance, but a few examples follow: a. You define sea-ice concentration (SIC) in the beginning of the abstract, but spell it out in other places. b. You defined sea-ice area (SIA) twice. 3. Some of your terminology is inconsistent throughout the manuscript. For example, sometimes you say "future SST and SIC", other times you say "projected SST and SIC", etc., which makes the manuscript difficult to follow. 4. There are a number of grammatical and spelling
errors, for example "The presence of SIC maps from futures AOGCM projections…" should read "The presence of SIC maps from future AOGCM projections…", please double-check the manuscript for grammar and spelling. 5. Figure 1 (right): It is difficult to determine which line is thick and which is thin, I suggest using a dashed line or adding more thickness. 6. Figure 3 caption: Where should the reader go to "see text"? 7. Figure 7: Including a key for the lines such as in Figure 4 would be helpful for clarity 8. Figure 8 and 9: The text refers to specific regions, for example the Weddell sea, but I'm not sure how to determine the regions from this Figure. 9. Figure A2 and A5: Including a key for the lines such as in Figure 4 would be helpful for clarity 10. Equation 1: As some of the parts of the equation refer to a climatological mean, and some to monthly data, adding in summations or "bar" notation would be very helpful. 11. Table 1 and resulting discussion: I may have missed something, but the labeling of this table confuses me, as well as the discussion in the text. In Section 3, below line 25, you that when comparing corrected RCP SST using the perfect model test and original SST from CNRM-CM5 RCP8.5 you obtain a null bias for the entire globe. Yet in this table you show CNRM-CM5 rcp8.5 – CNRM-CM5 hist has a mean difference of +3.04 degrees C. I assume something is written incorrectly here, but I'm not sure what. In addition, this table is referenced in only in Section 3.1.2, which references the IPSL-CM5A-LR data. I'm confused why you're changing models here. 12. As SIC is bias corrected independently of SSTA as noted in the first sentence of Section 3.3, this should also be mentioned somewhere in the methods section, providing context for the examination of physical consistency in Section 3.3.
* * *

---

## Author Comment (AC1) · 2 Mar 2018

The authors thank the referee for accepting to review the paper and for the generally constructive remarks aiming at the improvement of the quality of the paper. The responses to the different comments are below :

General comments This paper proposes a way to evaluate bias-correction methods for SST and SIC for future climate projections, using a perfect model approach and a real-case application. There has clearly been a large amount of work in this study and this is clear when reading the paper. The analysis is thorough and the discussion honest, with the main caveats being highlighted and explained (at least, an explanation is proposed). The conclusion is clear and includes potential other methods to investigate.

However, the presentation of methods and results might be a little bit confusing, given the amount of data.

1. Some extra introductory sentences explaining the point of using a perfect model approach would be welcome, as it might not be obvious to a reader that is not a specialist but wants to learn more. This could be done in section 2.4, where the description of the evaluation method (which is a main point of the paper) is a bit short. More generally, it would be interesting to provide some examples of the use of a perfect model approach in the literature.

Authors' response: "The reviewer is right to state that some reader might not be familiar with the perfect model approach. We therefore added the following sentences that should explain the approach in a nutshell and refer to some examples: "A perfect model approach usually consists of using model data as a substitute for observations, and trying to predict projected model data from that model; this prediction can then be evaluated against the available model projections (e.g., Hawkins et al., 2011). In the real world, as observations of future climate are obviously not yet available, an equivalent approach is impossible if one cannot wait long enough for the future to become reality. Another type of perfect model approach are "Big Brother" experiments for evaluating downscaling techniques. In such studies, high-resolution model output is degraded in resolution and downscaling methods are then applied to these low-resolution data. The resulting synthetic high-resolution fields are then compared to the original high-resolution output (e.g. de Elía et al., 2002). Here, we consider SST and SIC..."

2. If the language is usually clear and understandable, the wording can be unusual and the authors are encouraged to have (another?) correction by a native speaker (which the reviewer is not. . .)

Authors' response: "We had a native speaker correct the revised version."

3. On a more specific point, one might wonder why were the GCMs CNRM-CM5 and
IPSL-CM5A-LR chosen? Was this a choice based on availability or were these models selected based on their respective performance for representing SST and SIC? It would be nice to have information on this point.

Authors response: "The search for suitable bias-corrections methods and their use was first motivated by the need to drive future scenarios climate experiments with atmosphere-only GCMs ARPEGE and LMDZ. Therefore, the work was started with SST and SIC coming from the corresponding coupled model of these two atmosphere-only GCMs. HadGEM-ES was added later in order to verify if the results obtained were reproduced with this model, but we acknowledge that no criterion based on their respective performances has been used to select these models rather than another one. However, the fact that the results are very close for the three models investigated gives us confidence in the fact that they are robust and independent from the AOGCM chosen as initial material."

Finally some caveats and issues are treated too lightly and would require a more thorough description and explanation (see specific comments). Overall, the proposed paper describes an interesting and detailed work that should be of interest to many users in the climate modelling community. I therefore propose this manuscript to be accepted after the minor changes described in this review document.

Specific comments.

P1 L12: the part about RCPs is not needed, isn't it? The sentence is a bit long

Authors response: "Comment taken into account, the RCP acronym and the corresponding reference is now introduced in the Data section. "Only the first ensemble members of the historical, and of the Representative Concentration Pathways (RCPs, Moss et al., 2010) 4.5 and 8.5 simulations have been considered".

L17 bracket missing somewhere

Authors response: "Ok, comment taken into account."

GMDD
L19: Would it be possible to have some other examples from the literature? Surely a list of 4 or 5 references should be easy to find

Authors responses: "Some additional references were add as examples of the considerable literature on the bias of CMIP5 models, especially on SST. References demonstrating the added value of bias-corrected SST have been included as well: "The absence of the Pacific cold tongue bias and the reduction of the double ITCZ problem in AMIP experiments with respect to the CMIP5 model experiments (Li et al., 2014) shows the importance of forcing atmospheric model by SST close to the observations. For instance, improvements in the modelling of the tropical cyclone activity in the Gulf of Mexico (Holland et al., 2010) and of summer precipitation in Mongolia (Sato et al., 2007) were obtained by bias-correcting SST and other AOGCM outputs before using them as forcing for RCMs."

L19 "For example, it has. . ." Authors response: "Ok, comment taken into account."

L20 the seasonal cycle and the trend Authors response: "Ok, comment taken into account."

P2 L24 describe "AMIP" as "Atmospheric Model Intercomparison Project" if it is not done any- where else Authors response: "Ok, comment taken into account."

P3 L25 Is there a reference for the Hann box filter? Why did you choose this filter?

Authors response: "The first reference to Hann function is in "Particular pairs of windows" in "The measurement of power spectra, from the point of view of communications engineering" by R.B. Blackman and J. Tukey, 1959. We have chosen Haan filter because it is the lightest filter amongst the commonly used box filter."

P5 L13 Any information on the number / proportion of GCMs that were dismissed? What does "poorly" mean for the selection process?

Authors response: "We built our library by selecting AOGCMs who have a reasonable representation of the sea-ice extent annual cycle, its maximum and minimum, in
present climate following the literature (e.g. Turner et al., 2013, Stroeve et al., 2012). For instance, our list for the "real-case" application of the method contains historical simulation and future scenarios of the following AOGCMsÅä: MIROC-ESM, EC-EARTH, NorESM1-M, CCSM4 and IPSL-CM5A."

L13 AOGCMs and remove" overly"

Authors response: Ok, comment taken into account.

P7 L12 "We assume that an ideal bias correction method should reproduce the same change in mean and variance between the observations and the estimated future SST and SIC as between the used coupled GCM historical simulation and the climate change experiment." That seems obvious but is there any reference regarding this issue? Is there any discussion among the scientific community?

Authors response: There is indeed debate about this issue and so far, probably no consensus. For the bias-correction of future scenarios, one usually makes the hypothesis that one can rely on the climate signal coming from a model even if this model has bias in the reproduction of present climate. There are indeed reasons to believe that model biases are time invariant (e.g. Maurer et al. ,2013 (www.hydrol-earth-syst-sci.net/17/2147/2013) although whether we should correct the climate change signal remains an open question (see Ehret et al, 2012 (https://www.hydrol-earth-syst-sci.net/16/3391/2012/)).

L23 What is the point of applying the perfect model approach for SST, as we use only "regular" bias correction? You highlight this issue, but you might want to shrink this section a bit.

Authors response: Indeed, the part on bias correction of SST is less novel. Following this remark and remarks of the second reviewer, the part on the methodology for the bias-correction of SST as well as the part on the perfect model test have been shrunk.

P8 Fig4 Are you sure about the color? There seems to be a very large initial bias be-
tween the obs and the historical simulation for North Atlantic, is that expected? Moreover the RCP4.5 looks quite cold compared to the corrected values. If this is correct, can you highlight and explain that in the text?

Author responseÂă"The colors are right. The North Atlantic is a region where coupled GCMs often exhibit large biases (usually cold biases) because of their poor skills in representing correctly the Atlantic Meridional Overturning Circulation (AMOC). This example is indeed another argument for the bias correction of SSTs.

L11 "methods" Authors response: Ok, comment taken into account.

L12 delete "in" Authors response: Ok, comment taken into account.Âă

P9 L9 This comment is valid for the whole paper, but is the use of "biases" valid when describing the results of the perfect model experiment? It is a bit confusing with the original bias that we are trying to correct. Again, if it has been used previously in the literature in that context, I'm ok, but maybe "difference" or "error" would be clearer, as it is a bias created by the method, and not a bias originally in the data

Authors response: Comment taken into account, the term "error" or mean "error" is now used throughout the text in order to make it less confusing.

P10 L12 "more or less" – can we find a more scientific term please? Authors response:  $\hat{A}\tilde{a}Ok$ , comment taken into account.

L14 "is easy to explain" – Is it? Can you develop, please? Authors response: Ok, comment taken into account, some explanations are given in the text : "The presence of such peaks is easy to explain by taking into account the structure of the LUT as i) for a given month, the SIC does not always increase monotonically with decreasing SST, ii) the discrete nature of LUT is not in favour of a continuous SIC frequency distribution."

L29 Should an ideal method apply the same statistical changes? It sounds right, but what about skewed distribution (precipitation) where the BC would change the distribution, therefore changing the distribution of changes? I think there is quite a discussion

GMDD
about that topic, so, if I agree with you, I would change to "We consider here that an ideal method. . ." Authors response: Ok, the sentence has been modified following your recommendation.

P11 All text – Would it be possible to have some correlation value in order to quantify the error among the different methods? Maybe a correlation coefficient, or the value of the minimum, maximum and mean error for each graph?

Authors response: Mean errors and root mean square errors for each graph are added on the plots. In the text, we now discuss the average mean error or average RMSE for every scenarios and for the Arctic and Antarctic combined in order to quantify and compare more objectively the errors between the three methods.

Fig8 and 9 It is difficult to see which point correspond to what – Maybe adding a letter to each of them to point to the region would help – Please try but it might make the figure impossible to read. It would be nice to be able to navigate alone within the points

Author response: We changed the legend of the figure so that we can now distinguish the different regions with the help of different colors. Different signs (crosses and circles) are used to distinguish scenarios from CNRM-CM5 and IPSL-CM5A-LR. The more important for these figures is first to distinguish the regions, then the models. The distinction between rcp4.5 and rcp8.5 is less essential for the interpretation of the results and the connections with the text.

GMDD

---

## Author Comment (AC2) · 2 Mar 2018

The authors thank the referee for accepting to review the paper and for the generally constructive remarks aiming at the improvement of the quality of the paper. The responses to the different comments are below:

This manuscript discusses bias correction of sea-surface temperature using the anomaly method and the quantile-quantile method, and bias correction of sea ice concentration using the look-up table method, the iterative relative anomaly method, and the analog method. These bias correction methodologies are evaluated using a perfect model test (i.e. evaluated using the given model as "observations") and a real-case application in which the bias correction methods are compared to observations.

It is assumed that ideal bias correction will reproduce changes in the mean and variance between observations and projected climate as between historical simulations and projected climate. The authors determine that the presented methods for bias correcting SST are reliable. The methods presented for sea-ice concentration are less reliable, however, the analog method showed promising results and improvement over other bias correction methods. Additionally, the authors provide an appendix with a proposed method to parameterize sea-ice thickness, with potential for use in climate modeling applications. I have a number of major and minor comments for the authors to address. Some of the manuscript was unclearly written, making the arguments difficult to follow. I also question the inclusion of SST bias correction evaluation. In regards to the review criteria, the manuscript does present relevant information that is related to modeling questions, particularly for sea-ice concentration, rendering it suitable for publication. However, much of the methodology for sea-surface temperature bias correction has been noted in other manuscripts.

My comments are below. General Comments: 1. While the presented results for SIC are novel and will be very helpful for future modeling studies, the presented results for SST are somewhat less of an advancement. SST bias correction has been studied previously. In fact, there is much less discussion surrounding SST bias correction, and the results are almost glossed over by the authors in comparison. While the results are helpful in a summary sense for an interested reader, the concept seems less novel. This section may be able to be reduced even more, or eliminated completely.

Authors response: "We agree that this part of the paper is less novel and that these issues have been addressed in previous papers. Its presence in the manuscript is justified by the need to highlight the consistency with the work done for sea-ice, and the consistency between the response for the two variables, and to show the possibility to generalize the evaluation methods. However, in order to avoid redundant results, and emphasize the parts of the paper that are innovative, some parts of the result section were cut and the presentation of the methods have been mostly sent to the

Appendix section".

2. The Appendix describes a methodology for parameterizing sea-ice thickness, which was noted in Section 4.3 as a strong influence. While you state that an in-detail evaluation of sea ice thickness prescription is beyond the scope of this paper, you evaluate and further refine one of the methods for parameterization in the Appendix. This seems like an important contribution to the field that has been studied comparatively less than, for example, SST bias correction methodologies. I'm concerned that this contribution will be lost due to its presence in supplementary material, and would potentially warrant a separate manuscript that delves more deeply into the topic.

Authors response: "In some way, the work done on sea-ice thickness is not entirely innovative either, as it was already presented by one of the author (Krinner, 1997) and used in another study (Krinner, 2010). The innovation here is that the parameterization is further refined with parameters set for the Arctic and the Antarctic and that the results are objectively evaluated with sea-ice thickness measurements which so far were seldom, particularly in the Southern Ocean. However, we think that the current material on this topic is not sufficient to deserve a separate manuscript and it seems complicated to delves more deeply into the topic far enough to be able to produce a second manuscript. However, in order to avoid this contribution to be lost and in order to improve the manuscript consistency, we introduced the work on sea-ice thickness in the main part of the paper."

3. I am curious why the CNRM-CM5, IPSL-CM5A-LR, and HadGEM-ES coupled GCM data were explicitly chosen for this study. In addition, you note that HadGEM-ES was used in Section 2.1 near line 25, but never mention results from this model.

Authors response: "The search for suitable bias-corrections methods and their use was first motivated by the need to drive future scenarios climate experiments with atmosphere-only GCMs ARPEGE and LMDZ. Therefore, the work was started with SST and SIC coming from the corresponding coupled model of the latter two

atmosphere-only GCMs. HadGEM-ES was added later, in order to verify if the results obtained were reproduced with this model, but we acknowledge that criterion based on model performances was used to select this model rather than another one. However, the fact that the results are very close for the three models investigated gives us some confidence in the fact that they are robust and independent from the AOGCM chosen as initial material. Results/figures with HadGEM-ES are not presented in order to limit the length of the paper, nevertheless some of the results with HadGEM-ES could be included in the appendix section for some transparency purposes."

4. I am also curious why you selected the given bias correction methods for SST and SIC, are these arguably the most popular methods in use? If so, it would be helpful to note this as a motivation for the work.

Authors response: "Absolute anomaly and quantile-quantile methods are likely amongst the most popular methods for bias-corrections, especially for SST. Absolute anomaly for SST and iterative relative anomaly method for SIC have been introduced and used by one of the co-author in a previous work (Krinner et al., 2008). Evaluating the Look-Up Table method was motivated by the fact that this method is, so far, the method recommended in the frame of the HighResMIP for the production of bias-corrected SIC boundary conditions for atmospheric models. In the light of our results, this should be changed in favour of the analog method in a near future. We developed and introduced the analog method as we weren't satisfied by the results for the bias correction of SIC with the first two methods."

5. Because you are using a perfect model test, can these results be generalized to other models, or are these results specific to the models used?

Authors response: "We applied the perfect model test to rcp4.5 and rcp8.5 scenarios from three AOGCM of the CMIP5 experiments, and the results are very similar for each scenarios .From the perfect model test perspective, the results are not dependent on the models used. Relative performances of the three bias correction methods in the

"real-case" application also corroborate the results from the perfect model experiment which gives us confidence in the fact that results are essentially not model-dependent."

6. The introduction could benefit from additional discussion on SST biases, as it is written the focus is on SIC biases.

Authors response: "Some additional references were add as examples of the considerable literature on the bias of CMIP5 models, especially on SST. References demonstrating the added value of bias-corrected SST have been included as well in the introduction : "The absence of the Pacific cold tongue bias and the reduction of the double ITCZ problem in AMIP experiments with respect to the CMIP5 model experiments (Li et al., 2014) shows the importance of forcing atmospheric model by SST close to the observations. For instance, improvements in the modelling of the tropical cyclone activity in the Gulf of Mexico (Holland et al., 2010) and of summer precipitation in Mongolia (Sato et al., 2007) were obtained by bias-correcting SST and other AOGCM outputs before using them as forcing for RCMs.""

7. Figure 6 and resulting discussion: How does one determine what is a "reasonable" and "very small" error? To me, these look like large errors overall, but perhaps they are reasonable and very small with respect to the relative anomaly method?

Authors response: "The use of terms such as "reasonable" or "very small" has been reduced. Now, mean errors and root mean square errors for each graph were added on the plots. In the text, we now discuss the average mean error or average RMSE for every scenarios and for the Arctic and Antarctic combined in order to quantify and compare more objectively the errors between the three methods."

8. In Section 4.2, page 18, last sentence on the page: Preferentially selecting output of reasonably "well behaving" AOGCMs is perhaps too simplistically stated here. There are a variety of issues in selecting which models are "well behaving". Though the following reference focuses on selecting models for regional hydrological studies, some of the general comments will still hold true for model selection: Brekke LD, Dettinger

MD, Maurer EP, Anderson M (2008) Significance of model credibility in estimating climate projection distributions for regional hydroclimatological risk assessments. Clim Change 89:371–394 . doi: 10.1007/s10584-007-9388-3

Authors response: "Indeed, the selection of "well behaving" models for climate change applications is a complex issue extremely dependent on the processes and the region of interest. Further in the general discussion, we highlight this issue and add two references dealing with it : "The selection of climate models based on their credibility for climate change scenario is a complex issue (Brekke et al, 2008; Baumberger et al., 2017), dependent on the purposes, the processes and the region of study. Whether the climate change signal should be corrected remains on open question (Ehret, 2012), even though there are good reasons to believe that model biases are time invariant (Maurer et al., 2013).". In the discussion for the bias correction of SIC, we make clear that in the light of our results, it is preferable to avoid to use AOGCMs that have large or persistent negative bias on sea-ice in present climate as initial material for the analog or the iterative, relative anomaly method."

9. Is the main result for SST bias correction that either method is appropriate for use due to your evaluation of the reliability of these methods? How does this result differ from other work on SST bias correction?

Authors response: "Following our evaluation, this is indeed the main result for SST bias correction methods. This can be a little surprising as one can expect that the quantile-quantile method is more appropriate to reproduce change in variance and correct biases that are quantile-specifics. For the absolute anomaly, the fact that we use the complete time series of the AOGCM scenario rather than the climatological mean allows for taking into account the projected change in variance present in this scenario. However, given the fact that the quantile-quantile method is widely applied for bias correction of climate variables and has proven to be appropriate, at least for variables that have no skewed distribution such as temperatures, we would recommend the use of the quantile-quantile method. This is however not the main point of the

paper."

Technical Comments: The following comments should be easy to address, but will substantially improve the readability of the manuscript.

1. Please confirm that all acronyms are clearly defined, I have not listed all instances, but a few examples follow: CMIP5, AOGCM, AMIP, PCDMI, AGCM, etc.

Authors response: "Ok, comment taken into account."

2. Please confirm that all acronyms are consistent throughout the manuscript. I have not listed all instance, but a few examples follow: a. You define sea-ice concentration (SIC) in the beginning of the abstract, but spell it out in other places. b. You defined sea-ice area (SIA) twice.

Authors response: "Ok, comment taken into account"

3. Some of your terminology is inconsistent throughout the manuscript. For example, sometimes you say "future SST and SIC", other times you say "projected SST and SIC", etc., which makes the manuscript difficult to follow.

Authors response: "Ok, comment taken into account"

4. There are a number of grammatical and spelling errors, for example "The presence of SIC maps from futures AOGCM projections. . ." should read "The presence of SIC maps from future AOGCM projections. . .", please double-check the manuscript for grammar and spelling.

Authors response: "Comment taken into account, we had the manuscript read by a native speaker."

5. Figure 1 (right): It is difficult to determine which line is thick and which is thin, I suggest using a dashed line or adding more thickness.

Author response: "Ok, comment taken into account"
[Figure]

6. Figure 3 caption: Where should the reader go to "see text"?

Authors response: "Comment taken into account, there is now reference towards the corresponding section."

7. Figure 7: Including a key for the lines such as in Figure 4 would be helpful for clarity

Authors response: "Ok, comment taken into account"

8. Figure 8 and 9: The text refers to specific regions, for example the Weddell sea, but I'm not sure how to determine the regions from this Figure.

Authors response: "We changed the legend of the figure so that we can now distinguish the different regions with the help of different colors. Different signs (crosses and circles) are used to distinguish scenarios from CNRM-CM5 from IPSL-CM5A-LR. The more important for these figures is first to distinguish the regions, then the models. The distinction between rcp4.5 and rcp8.5 is less essential for the interpretation of the results and the connections with the text."

9. Figure A2 and A5: Including a key for the lines such as in Figure 4 would be helpful for clarity

Authors response: "Ok, comment taken into account."

10.Equation 1: As some of the parts of the equation refer to a climatological mean, and some to monthly data, adding in summations or "bar" notation would be very helpful.

Authors response : "Ok, comment taken into account."

11. Table 1 and resulting discussion: I may have missed something, but the labeling of this table confuses me, as well as the discussion in the text. In Section 3, below line 25, you that when comparing corrected RCP SST using the perfect model test and original SST from CNRM-CM5 RCP8.5 you obtain a null bias for the entire globe. Yet in this table you show CNRM-CM5 rcp8.5 – CNRM-CM5 hist has a mean difference of +3.04 degrees C. I assume something is written incorrectly here, but I'm not sure

what. In addition, this table is referenced in only in Section 3.1.2, which references the IPSL-CM5A-LR data. I'm confused why you're changing models here.

Authors response: "The goal of the real case application is to show that the mean and standard deviation shift due to the climate change is similar between the observations in the historical periods and the bias corrected scenario than between the historical simulation and the original scenario of the AOGCM used as initial material. Perhaps, the term mean difference was confusing here, we propose to replace it by change in mean and change in standard deviation."

12. As SIC is bias corrected independently of SSTA as noted in the first sentence of Section 3.3, this should also be mentioned somewhere in the methods section, providing context for the examination of physical consistency in Section 3.3.

Authors response: "We now refer to the examination of the physical consistency analysis in the introduction and the method section of the paper : "As SST and SIC are bias-corrected separately, section~\ref{sec3.3} presents a few considerations about SST and SIC consistency after performing bias corrections. The effects of the corrections applied \textit{a posteriori} in order to ensure the physical consistency between the two variables are evaluated within the framework of the perfect model test."

---

## Referee Report (RR1)

The authors investigate effects of different bias-correction methods of ocean surface conditions of CMIP5 models. They discusses the anomaly and quantile-quantile well-know methods for Sea Surface Temperature (SST) corrected existing (look-up table and relative anomaly) and new (analog) methods for Sea-Ice Concentration (SIC). A particular attention is paid to the consistency between SST and SIC fields. Furthermore, a method for correcting Sea-Ice Thickness is also (re)presented and improved by taking into account different parameters for North and South hemisphere. Results of each method are evaluated against a perfect model test and real-case application. They show that the analog method and the proposed method to parameterize Sea-Ice Thickness respectively improve the bias correction of SIC and SIT.

The study is well designed. It presents detailed analysis, a fair and honest discussion about bias correction methods highlighting their advantages and the hypothesis on which they rest. Compared to their first version the authors have strongly improved their manuscript by taken into account reviewer comments even if some points could be improved. The authors can find my comments and stylistic suggestions hereunder.

**General comment**

Like the reviewer 2, I am in a way surprise that results with HadGEM-ES are never mentioned while using a third model with almost the same results could increise their confidence. I understand that authors already have a lot a results to present but maybe could they add some results and comparison in supplementary file/materials attached to their article?

**Point comments**

P1, L33 « high-resolution atmospheric global circulation models »
Although some CMIP6 models has performed simulations at higher resolutions than CMIP5 simulations (25-50km for the high resolution model intercomparison project, Haarsma *et al.*, 2016) ; which is an important improvement in comparison to CMIP5, 150km), this range of "high" horizontal resolution appears to be as the lowest RCM resolutions in the CORDEX project so that the therm « high-resolution » should (at least) also qualified « Regional Climate Models » (p1,L32).

P3,L67. Are the results influenced by the selection (and the number) of these sectors? Have the authors tested an other definition of these sectors? Are these sectors only defined by their geographical location or also by the recent (local) SIC trends observed in the Arctic and the Southern Oceans? See also the comment in Figures and Tables section.

P4,L9. I support the fact that authors have dismissed from their library the CMIP5 AOGCMs that poorly represent the sea-ice annual cycle in present-day climate. However, as already remarked by F. Gallo (Reviewer 1), the selection process is unclear. At this stage of the manuscript, one could wonder how have these "good" models been selected by the authors or how did the authors treat models that are correct for the Arctic ocean and poor for the Southern ocean as for instance MIROC5 (Shu *et al.*, 2015; Turner *et al.*, 2013).
The authors' answer to R1 could appear in the final version of the manuscript as two reviewers made a comment on it. I also suggest to specify again at the AOGCMs that have been used (or at least to make a reference to section data at P4,L9).

**Figures and Tables**

P3, L67. The authors should consider to add a map (or at least a table) in supplementary material showing (listing) the defined sectors used in the analog method.

Figure 3 (P6). Unless I'm mistaken, the reference to Figure 3 is missing in your text.

Figure 5 Maybe this figure could be placed in supplement as it has less scientific interest?

Figure 8 and 9 Is it interesting to differentiate the RCP scenarios? As your results do not depend on the RCP scenarios, maybe it is not interesting to differentiate it but two same symbols (circles or crosses) with the same colors and (slight) different meanings are a little bit confusing. However, I note that the authors have strongly increase the clarity of theses figures in comparison with their first version.

Figure 11 and Figure 14 Could the authors consider to merge the two figures as it will make the comparison easier between the general and Arctic-specific parameter results.

Figure 12 and 15; Figure 13 and 16 I also suggest to the authors to merge the corresponding figures of SIT with general and specific parameters leading to easier comparisons. Maybe could you make a panel with two ranks and still 3 columns "Observed, Error General Parameter, Error specific parameter (for instance Observed, March; Error(General Parameter), March; Error(Specific Parameter) March).

**Stylistics comments**

"sea ice" versus "sea-ice", please check the consistency of those spellings in all your manuscript (vs sea-ice); for instance "sea-ice thickness" (P1,L30) versus "sea ice thickness" (P5,L4)

P1, L36 "Sea Surface Temperature (SST), Sea Ice Concentration (SIC), Sea-Ice thickness (SIT)" maybe you could consider "Sea-Ice Thickness (SIT)" to be consistent with the first two terms (see also P5L8 and comment for P3,L25);

P2, L11 "by an atmospheric model" I would suggest to specify the type of the atmospheric model as follows 'by an atmospheric global circulation model"

P3, L20 "in this method,the assumption", please add a space "in this method, the assumption"

P3, L35 "sea-ice area (SIA)", following comments on P1,L36 "Sea-Ice Area (SIA)"

P3,L40 "With respect to the method introduced in Krinner et al. (2008), we introduce" I suggest "With respect to the method described by Krinner et al. (2008), we introduce"

P6, L10 "In figure 4 (bottom), we can see the large cold bias of the AOGCM..." Maybe you could modify as follows "Figure 4 (bottom) also show the large cold bias of IPSL-CAM5A-LR..."

P11, L32 "in the tropics,the West African" a space is missing

P12, L23 "In the perfect model test, we have seen that the LUT method shows some reduced errors over most regions (Figure 6). However, we have seen that the frequency distribution of future SIC..." I would suggest to reformulate as follows "The perfect model test pointed out that the LUT

method shows some reduced errors over most regions (Figure 6). However, the frequency distribution of future SIC..”

P12, L34 “the use of SST as a proxy for SIC...” maybe “using SST as a proxy for SIC...”

P12, L51 “the decrease of sea-ice” please change with “the decrease in sea-ice”

P14, L32* “of the sea-ice in the Antarctic,...” maybe “of the Antarctic sea-ice,...”

P14, L* “The fact that both methods over-estimate the decrease in sea ice mainly for CNRM-CM5 scenarios is to be linked to the fact that the historical simulation of this AOGCM shows some considerable negative biases for the sea-ice in the Weddel Sea with respect to the observations”
I suggest  “The fact that both methods over-estimate the decrease in sea ice mainly for CNRM-CM5 scenarios is to be linked to some considerable negative biases of its historical simulation for the sea-ice in the Weddel Sea with respect to the observations.”

P14, L* “4.3 Sea-ice Thickness” see first comment about P1,L36

P14,L* “The Central Arctic SITs results” modify by “The Central Artic SIT result”

*(\*in the version of your manuscript that I read, line numbers seem to become uncertain starting from page 14. I hope you will be able to find where I suggest you to modify something. My apologies for the inconvenience.)*

**References**

Shu, Q., Song, Z., and Qiao, F.: Assessment of sea ice simulations in the CMIP5 models, The Cryosphere, 9, 399-409, https://doi.org/10.5194/tc-9-399-2015, 2015.

Turner, J., Bracegirdle, T. J., Phillips, T., Marshall, G. J., & Scott Hosking, J. An initial assessment of antarctic sea ice extent in the CMIP5 models. Journal of Climate, 26(5), 1473–1484. https://doi.org/10.1175/JCLI-D-12-00068.1, 2015.

---

## Author Response (AR2)

**Assessing bias-corrections of oceanic surface conditions for atmospheric models**
**Beaumet J., Krinner G., Déqué M., Haarsma R., Li. L.**

**We thanks the anonymous referee #3 for constructive remarks aiming at the improvement of the final version of the manuscript and referee #2 for accepting the manuscript as such.**

**gmd-2017-247-referee-report-2 (09-08-2018): Author's response**

**General comment**
Like the reviewer 2, I am in a way surprise that results with HadGEM-ES are never mentioned while using a third model with almost the same results could increase their confidence. I understand that authors already have a lot a results to present but maybe could they add some results and comparison in supplementary file/materials attached to their article?

Authors response : " The present study has been submitted almost one year ago and all we have left "in store" are results of the perfect model test application for HadGEM2-ES RCP4.5 and RCP8.5 scenarios with the Look-Up Table and the analog method for SIC. Generating a whole set of new results for "real-case application" is beyond the scope of minor revisions and substantial changes in the manuscript would require a new round of revision. Besides this, the article already contain a rather large amount of results and figures. However, in order to increase the confidence in the results, we added a figure (spatial distribution of the errors) for the results with HadGEM2-ES model. The results are extremely similar to those obtained with CNRM-CM5 and IPSL-CAM5A-LR models which increase the confidence in the independence of the results from the model used. Moreover, since the submission of the present study, the analog method for SIC has been successfully applied to NorESM1-M, MIROC-ESM, BNU-ESM, EC-EARTH and CanESM2 models. The reconstruction of the climate change signals originally present in the AOGCM scenario has been verified for the Southern Oceans. The results have been used to drive ARPEGE and LMDZ atmosphere-only GCM with stretched grid configuration over Antarctica and yielded extremely satisfactory results."

**Point comments**
P1, L33 « high-resolution atmospheric global circulation models »
Although some CMIP6 models has performed simulations at higher resolutions than CMIP5 simulations (25-50km for the high resolution model intercomparison project, Haarsma et al., 2016) ; which is an important improvement in comparison to CMIP5, 150km), this range of "high" horizontal resolution appears to be as the lowest RCM resolutions in the CORDEX project so that the therm « high-resolution » should (at least) also qualified « Regional Climate Models » (p1,L32).

Authors response : "Ok, we propose to qualify RCMs with the term "(very) high-resolution" and "high-resolution" for AGCMs."

P3,L67. Are the results influenced by the selection (and the number) of these sectors? Have the authors tested an other definition of these sectors? Are these sectors only defined by their geographical location or also by the recent (local) SIC trends observed in the Arctic and the Southern Oceans? See also the comment in Figures and Tables section.

Authors response : " We have not tested the influence of the number of sectors chosen neither different methods for their definitions. We acknowledge that this could have been investigated, although, this effort is beyond the scope of a minor revision. However, we have tested different values for the reference distance ($d^r$ in equation 4, P4L49)) and other values for the exponent above

(d$_{(i,s)}$/d$_r$). The values kept optimizes the reproduction of the climate changes trends in the center of each sector while ensuring smooth transition of correction sea-ice fields between each sectors. "

P4,L9. I support the fact that authors have dismissed from their library the CMIP5 AOGCMs that poorly represent the sea-ice annual cycle in present-day climate. However, as already remarked by F. Gallo (Reviewer 1), the selection process is unclear. At this stage of the manuscript, one could wonder how have these "good" models been selected by the authors or how did the authors treat models that are correct for the Arctic ocean and poor for the Southern ocean as for instance MIROC5 (Shu et al., 2015; Turner et al., 2013).

The authors' answer to R1 could appear in the final version of the manuscript as two reviewers made a comment on it. I also suggest to specify again at the AOGCMs that have been used (or at least to make a reference to section data at P4,L9).

Authors response : "The response to R1 has been added to the manuscript and the list of AOGCMs used to build the library of the analog method is presented in the supplementary material, with a reference towards it in the main part of the article. Here, we acknowledge that our selection of models could have been performed in a more rigorous way and that other models could have been selected as "good models", while the use of the MIROC-ESM could have been avoided due to its underestimation of SIE in the Arctic.. However, the use of a non-exhaustive list of model performing "well" for sea-ice in the Arctic and in the Antarctic is probably non detrimental to the performance of the method. The goal of the selection process is to avoid keeping models that have unphysical or aberrant sea-ice concentration spatial distribution in the Arctic or the Antarctic region in the historical period, which is not the case for the models selected. "

**Figures and Tables**

P3, L67. The authors should consider to add a map (or at least a table) in supplementary material showing (listing) the defined sectors used in the analog method.

Authors response : "Comment taken into account, a map has been added in the supplementary material section".

Figure 3 (P6). Unless I'm mistaken, the reference to Figure 3 is missing in your text.

Authors response "Comment taken into account and corrected, a reference to the figure has been in the text, at the end of the introduction section (P2 L36) ".

Figure 5 Maybe this figure could be placed in supplement as it has less scientific interest?

Authors response "Comment taken into account and corrected, the figure has been put in the supplementary material".

Figure 8 and 9 Is it interesting to differentiate the RCP scenarios? As your results do not depend on the RCP scenarios, maybe it is not interesting to differentiate it but two same symbols (circles or crosses) with the same colors and (slight) different meanings are a little bit confusing. However, I note that the authors have strongly increase the clarity of theses figures in comparison with their first version.

Authors response "Indeed, here again, the results do not depend much from the model considered, we removed therefore the uses of crosses to differentiate IPSL-CM5A-LR from CNRM-CM5 scenarios in order to make it less confusing.".

Figure 11 and Figure 14 Could the authors consider to merge the two figures as it will make the comparison easier between the general and Arctic-specific parameter results.

Authors response "Ok, the figures have been gathered in a 2 sub-figure pannel to facilitate the comparisons".

Figure 12 and 15; Figure 13 and 16 I also suggest to the authors to merge the corresponding figures of SIT with general and specific parameters leading to easier comparisons. Maybe could you make a panel with two ranks and still 3 columns "Observed, Error General Parameter, Error specific parameter (for instance Observed, March; Error(General Parameter), March; Error(Specific Parameter) March).

Authors response "Ok, the figures have been gathered in a 2 sub-figure panel to facilitate the comparisons".

**Stylistics comments**

"sea ice" versus "sea-ice", please check the consistency of those spellings in all your manuscript (vs sea-ice); for instance "sea-ice thickness" (P1,L30) versus "sea ice thickness" (P5,L4)

Authors response "Ok, comment taken into account and corrected, the spelling "Sea-Ice is consistently kept along all the manuscript".

P1, L36 "Sea Surface Temperature (SST), Sea Ice Concentration (SIC), Sea-Ice thickness (SIT)" maybe you could consider "Sea-Ice Thickness (SIT)" to be consistent with the first two terms (see also P5L8 and comment for P3,L25);

Authors response "Ok, comment taken into account and corrected, the spelling "Sea-Ice is now consistently kept along all the manuscript".

P2, L11 "by an atmospheric model" I would suggest to specify the type of the atmospheric model as follows 'by an atmospheric global circulation model"

Authors response "Ok, comment taken into account and corrected".

P3, L20 "in this method,the assumption", please add a space "in this method, the assumption"

Authors response "Ok, comment taken into account and corrected".

P3, L35 "sea-ice area (SIA)", following comments on P1,L36 "Sea-Ice Area (SIA)"

Authors response "Ok, comment taken into account and corrected".

P3,L40 "With respect to the method introduced in Krinner et al. (2008), we introduce" I suggest "With respect to the method described by Krinner et al. (2008), we introduce"

Authors response "Ok, comment taken into account and corrected".

P6, L10 "In figure 4 (bottom), we can see the large cold bias of the AOGCM..." Maybe you could modify as follows "Figure 4 (bottom) also show the large cold bias of IPSL-CAM5A-LR..."

Authors response "Ok, comment taken into account and corrected".

P11, L32 "in the tropics,the West African" a space is missing

Authors response "Ok, comment taken into account and corrected".

[revised manuscript text omitted]